# "Moving from one environment to another, it doesn't automatically change everything". Exploring the transnational experience of Asian-born gay and bisexual men who have sex with men newly arrived in Australia

Tiffany R. Phillips[1,2]*, Nicholas Medland[1,2,3], Eric P. F. Chow[1,2,4], Kate Maddaford[1], Rebecca Wigan[1], Christopher K. Fairley[1,2], Jason J. Ong[1,2‡], Jade E. Bilardi[1,2,5‡]

1 Melbourne Sexual Health Centre, Alfred Health, Melbourne, VIC, Australia, 2 Central Clinical School, Monash University, Melbourne, VIC, Australia, 3 The Kirby Institute, University of New South Wales, Sydney, Australia, 4 Centre for Epidemiology and Biostatistics, Melbourne School of Population and Global Health, The University of Melbourne, Melbourne, VIC, Australia, 5 Department of General Practice, University of Melbourne, VIC, Australia

‡ These authors are joint senior authors on this work
* TPhillips@mshc.org.au

## Abstract

Asian-born gay, bisexual and other men who have sex with men (gbMSM) who are newly arrived in Australia are at a higher risk of acquiring HIV than Australian-born gbMSM. We used a social constructionist framework to explore HIV knowledge and prevention strategies used by newly-arrived Asian-born gbMSM. Twenty four Asian-born gbMSM, aged 20–34 years, attending Melbourne Sexual Health Centre, who arrived in Australia in the preceding five years, participated in semi-structured, face-to-face interviews. Interviews were recorded, transcribed verbatim and analysed thematically. Participants described hiding their sexual identities in their country of origin, particularly from family members, due to fear of judgement and discrimination resulting from exposure to sexual identity and HIV related stigma in their countries of origin, although some were open to friends. Despite feeling more sexual freedom and acceptance in Australia, many were still not forthcoming with their sexual identity due to internalised feelings of stigma and shame. Exposure to stigma in their country of origin led many to report anxiety around HIV testing in Australia due to a fear of testing positive. Some described experiencing racism and lack of acceptance in the gay community in Australia, particularly on dating apps. Fear of discrimination and judgement about their sexual identity can have a significant impact on Asian-born gbMSM living in Australia, particularly in terms of social connectedness. Additionally, HIV-related stigma can contribute to anxieties around HIV testing. Our data highlights the potential discrimination Asian-born gbMSM face in Australia, which has implications for social connectedness, particularly with regard to LGBTQI communities and HIV testing practices. Future studies should determine effective strategies to reduce sexual identity and HIV-related stigma in newly-arrived Asian-born gbMSM.

**Data Availability Statement:** Due to the small sample size and the interview transcripts containing sensitive and potentially identifying information, the ethics committee have not approved public release of this type of data. Interested researchers may contact Emily Bingle at the Alfred Hospital Ethics Committee if they would like access to the data: research@alfred.org.au, quoting project 222/19.

**Funding:** This study was funded by Gilead Pharmaceuticals research fellowship. The funders had no role in study design, data collection and analysis, decision to publish, or preparation of the manuscript. https://gileadfellowship.com.au/.

**Competing interests:** NM, EPFC and JJO have received a research grant from Gilead Pharmaceuticals to conduct this investigator-initiated study. Gilead Pharmaceuticals had no input in the research plan, recruitment, analysis or decision to publish the paper. EPFC is supported by an Australian National Health and Medical Research Council (NHMRC) Emerging Leadership Investigator Grant (GNT1172873). JJO is supported by an Australian NHMRC Early Career Fellowship Grant (APP1104781). JEB is supported by an Australian Research Council Discovery Early Career Researcher Award (DECRA) Fellowship (DE200100049). All other authors have no conflicts of interest to declare. This does not alter our adherence to PLOS ONE policies on sharing data and materials.

# Introduction

There has been a decline in new HIV diagnoses since 2016 among gay, bisexual and other men who have sex with men (gbMSM) born in Australia [1,2]. The proportion of HIV diagnoses among Asian-born gbMSM living in Australia have increased significantly, from 9% in 2008 to 23% of all new male-to-male HIV diagnoses in 2017 [2]. By 2017, gbMSM born in Asia were more than four times as likely to be diagnosed with HIV infection than other gbMSM living in Australia (1.6% vs 0.4%, p<0.001) [3].

The overall aim of this study was to explore HIV knowledge and prevention strategies used and preferred among newly-arrived Asian-born gbMSM, the results of which are pending publication in a separate paper. In exploring this, it became evident men experienced a trans-national duality of lived experiences as a gbMSM in their country of origin and in Australia, strongly influenced by their exposure to deeply embedded societal and cultural beliefs around same-sex relationships and HIV in their country of origin. Despite feeling greater acceptance and sexual freedom in Australia, the stigma and shame around their sexual identity and HIV in their country of origin, endured and influenced their living experience in Australia, where some also faced other forms of sexual and racial minority discrimination.

The minority stress model explains the effects of minority sexual identity on mental health and posits that marginalised groups experience more social stress than non-marginalised groups [4]. Asian-born gbMSM who are migrants or temporary visa holders (e.g. students or those migrating permanently) in Australia are socially categorized in marginal groups not only due to their sexual identity, but also their ethnicity and migration status; an intersectionality that can contribute to HIV-related stigma, and, in turn, have effects on HIV-related health behaviours [5]. A previous systematic review of migrant minority gbMSM in the USA and Europe found varying HIV risk profiles in migrant gbMSM depending on their ethnicity, country of origin, and current location and highlighted the importance of viewing HIV risk in the context of migration [6]. In order to understand the broader range of inequalities including racism, discrimination and socio-economic status that may be contributing to HIV vulnerability in Asian-born gbMSM living in Australia, it is essential to use an intersectional approach that examines culture, integration into the host community, and health of the individual [7].

It is well documented that HIV-related stigma and shame pose significant barriers for accessing HIV testing [8,9]. HIV-related stigma continues to impede efforts to reduce HIV transmission in Australia as it presents a major barrier for accessing care [10]. Additionally, sexual identity-related stigma contributes to mental distress [11,12] and can lead to concealment of sexual identity [13,14]. Sexual identity concealment, in turn, is associated with worse mental health outcomes [15], a decreased sense of belonging, and social isolation [4,16]. In Australia, Asian-born gbMSM are more likely to have advanced HIV upon diagnosis but to report fewer numbers of male sexual partners compared to Australian-born gbMSM [17]. This may indicate an underlying stigma related to HIV and sexual identity, particularly given the reported prevalence of such stigma in several Asian countries [18–22].

To better understand the underlying beliefs around HIV and, in turn, the possible contributors to HIV vulnerability in newly-arrived Asian-born gbMSM, it is important to examine the complex relationship of social and cultural values attributed to HIV in their countries of origin as well as their experience in sexual and ethnic minority communities within their countries of origin and Australia [23]. There are limited data on the experience of Asian-born gbMSM migrants in Australia [24]. With this population continuing to be at a higher risk of HIV than Australian-born gbMSM [3] it is important to understand the intersectionality of their experience and their experiences of HIV-related stigma.

## Materials and methods

This study has been reported in accordance with the Consolidated Criteria for reporting qualitative research (COREQ) guidelines [25].

### Ethics statement

Ethical approval was obtained from the Alfred Hospital Ethics Committee, Victoria, Australia (222/19) on the 30th April 2019.

### Theoretical framework

The framework for this study was informed by a social constructionist approach. According to this approach, an individual's perceptions of their reality and meaning they give to phenomena (beliefs, values, experiences) are shaped by the social and cultural norms in which they live [26]. From this viewpoint, each man's experience or 'reality' of living as an Asian-born gbMSM in their country of origin and Australia will differ and what is important is not the accuracy of their accounts but rather the insights they provide into their realities or lived experience. The way an illness or condition is viewed or responded to in a culture or society can greatly impact or influence the lived experience or perceived reality of individuals, especially if the illness is stigmatised [27]. Given the considerable negative stigma still surrounding both same-sex attraction and HIV in many Asian countries, it was anticipated men's views and experiences of living as a gbMSM in Australia would be influenced by their country of origin's societal values and beliefs.

### Method, research team and reflexivity

Semi-structured interviews were chosen for this study as they allowed men to share their lived experience and personal reality of being a gbMSM. The interview schedule was jointly designed by a majority of the research team, including NM (FAChSHM, PhD), a male clinical epidemiologist and consultant HIV physician; EPFC (PhD) a male epidemiologist with several years' experience in sexual health; JJO (FAChSHM, PhD), a male sexual health physician and researcher with a special interest in increasing access to sexual health services to marginalised populations; JEB (PhD), a female senior research fellow with a doctorate in public health who specialises in social research in the area of sexual and reproductive health; and TRP (PhD), a female research fellow with several years' experience working in sexual health. NM and JO particularly had extensive quantitative research experience describing trends in HIV diagnoses within newly-arrived Asian-born gbMSM. The team members have combined decades of experience in this research area, have diverse sexual identities and cultural backgrounds, including two members of Asian ethnicity. Thus, the nature of the questions chosen were influenced by the clinical and cultural experiences of the researchers themselves, which in turn will have impacted on the meanings derived from the interviews.

Interviews were conducted by TRP. Participants had no prior relationship with TRP and were informed that the study was being undertaken to understand their experiences around HIV testing and preferences for HIV prevention strategies in light of rising rates of HIV diagnoses in this population.

### Recruitment

Participants were recruited from the Melbourne Sexual Health Centre (MSHC) between 8th May 2019 and 23rd December 2019. MSHC is the largest public sexual health centre located in the State of Victoria, Australia, providing approximately 60,000 consultations in 2019 [28]. All

clients attending MSHC are invited to complete a computer-assisted self-interview (CASI) on arrival, which collects demographic and sexual practice information. Men were purposively invited to participate. During the study period, all clients who self-identified on CASI as a male aged 18 years or older, reported having sex with men in the last 12 months, were HIV negative, and born in Asia, were automatically shown an invitation on CASI to take part in an interview. The invitation on CASI contained additional eligibility criteria which stated that participants needed to have been in Australia less than five years and that they needed to have been born in an Asian country, defined as: Bangladesh, Bhutan, Brunei, Cambodia, China, East Timor, Hong Kong, India, Indonesia, Japan, North Korea, South Korea, Laos, Macau, Malaysia, Maldives, Mongolia, Myanmar, Nepal, Pakistan, Philippines, Singapore, Sri Lanka, Taiwan, Thailand or Vietnam. Interested men indicated a preference for contact from the study team either via email or phone call. A research nurse (KM or RW) then contacted men using their preferred method of contact, confirmed eligibility, emailed them the plain language statement and consent form, and scheduled a time for them to attend MSHC for an interview.

## Data collection

All interviews were conducted face to face, in English and in a private room at MSHC. After reviewing the plain language statement, written informed consent was obtained. Interviews were digitally recorded. To begin, participants were asked nine structured questions on demographic and sexual practices, followed by a series of open-ended questions around their experience of living as a gbMSM in their country of origin, their experience of living as a gbMSM in Australia, their HIV and STI knowledge, their HIV prevention strategies and any changes in sexual behaviour or HIV prevention strategies since coming to Australia (see Table 1). This paper reports findings on the transnational experience of living as a gbMSM in their country

**Table 1. Sampling framework, eligibility criteria and interview schedule topics.**

| **Sampling framework** |
| --- |
| • Gay, bisexual and other men who have sex with men (gbMSM) |
| • Recently arrived in Australia (less than five years ago) from an Asian country |
| • Clients of Melbourne Sexual Health Centre |
| • HIV negative |
| *Revised during data collection to include*: |
| • More men from countries outside of China |
| **Eligibility Criteria** |
| • Male |
| • Aged 18 years or older |
| • Good understanding of English |
| **Interview schedule topics** |
| • Experience of being gbMSM in country of origin |
| • Experience of being gbMSM in Australia |
| • HIV and STI knowledge* |
| • Preferred HIV prevention strategies* |
| *Revised during data collection to include additional questions* |
| • How HIV is viewed in participant's country of origin |
| • Changes in sexual behaviour or prevention strategy since coming to Australia* |
| • Participant's perceptions of HIV risk* |

*Study results for these topics are reported in a separate paper.

of origin and Australia, including societal perceptions of HIV and personal experiences of HIV testing. Findings related to men's HIV and STI knowledge and HIV prevention strategies are reported in a separate paper (as yet unpublished). Field notes were written by TRP immediately following each interview to provide contextual detail. All interviews were transcribed verbatim for thematic analysis. Following the interview, participants received a $AUD30 voucher as reimbursement for their time and travel, as well as an information packet containing information on HIV prevention (including PrEP), health resources, service providers, and information about a community-based Victorian lesbian, gay, bisexual, trans, queer, and intersex (LGBTQI) health organisation, Thorne Harbor Health (formerly the Victorian AIDS Council). Upon request, two participants were assisted in scheduling appointments with on-site counsellors. Participants were informed that they could request a copy of their interview transcripts for member checking, however all participants declined.

### Data analysis

The research team met regularly to discuss the findings from the study, which involved reflecting on the research methodology and identifying areas of improvement within the interview structure as well as reflexively examining and challenging our underlying perspectives about the research participant's attitudes and cultural influences [29].

The interview schedule was revised after two interviews were completed to add additional questions around how HIV was viewed in the participant's country of origin and how their sexual practices might have changed over time. After seven interviews were completed the interview schedule was further refined to include questions about participant's perceptions of HIV risk (see Table 1). The sampling framework was refined after approximately two-thirds of the interviews were completed to broaden the countries of origin in our study population, specifically restricting recruitment of men from China to ensure this country was not over-represented in the data collection. After 24 interviews were completed, TRP and JEB reviewed the transcripts and discussed results, at which time it was decided that data saturation had been met and no further interviews were required.

Interview transcripts were reviewed for thematic analysis [30] by TRP using both a deductive (whereby themes were identified by drawing on previous literature and the interview schedule) and inductive approach (by examining emergent and recurrent themes arising independently from the data). Transcripts were imported into QSR International's NVivo 12 software for data management. Thematic analysis commenced with TRP reading each transcript and coding responses. Codes were subsequently grouped into potential themes and subthemes, and reviewed, refined and compared for similarities and differences. A subset of transcripts were independently read and analysed by JEB to cross check coding and reduce bias. Consensus was met between TRP and JEB on coding and themes with no differences in data interpretation identified between the two researchers.

Descriptive analyses of demographic information were conducted using Stata (version 14.0, College Station, TX: StataCorp LP).

### Results

A total of 117 men registered their interest in the study, of whom 97 fit the eligibility criteria. There were 38 men who scheduled an interview and of those, 14 did not attend or reschedule. Twenty four men completed an interview before data saturation was met. Interviews took 42 minutes on average [range 25 to 57 minutes]. Participant demographics are reported in Table 2.

Two main themes and nine subthemes emerged from the data:

**Table 2. Participant demographics.**

| | n or median [range] |
|---|---|
| **Number of participants** | 24 |
| **Age (years)** | 27 [20–34] |
| 20–24 | 6 |
| 25–29 | 12 |
| 30–34 | 6 |
| **Sexual Identity*** | |
| Gay | 21 |
| Bisexual | 1 |
| In between gay and bisexual | 1 |
| Queer | 1 |
| **Country of origin and ethnicity†** | |
| China | 5 |
| Indonesia | 3 |
| Balinese-Javanese | 1 |
| Javanese | 1 |
| Native Indonesian | 1 |
| Malaysia | 3 |
| Malay | 1 |
| Sino-Kadazan | |
| None given | 1 |
| Laos | 2 |
| Philippines | 2 |
| Singapore | 2 |
| Malay | 1 |
| Chinese | 1 |
| Taiwan | 2 |
| India | 1 |
| Pakistan | 1 |
| Sri Lanka | 1 |
| Sinhalese | 1 |
| Thailand | 1 |
| Vietnam | 1 |
| **Religious affiliation** | |
| Muslim | 6 |
| Buddhist | 6 |
| Catholic | 3 |
| Catholic and Buddhist | 2 |
| No Religion | 4 |
| Did not disclose or was not asked about religion | 3 |
| **Length of time in Australia** | |
| <12 months | 5 |
| 1 year | 6 |
| 2 years | 3 |
| 3 years | 5 |
| 4 years | 5 |
| **Occupation** | |
| Postgraduate student | 10 |

(*Continued*)

**Table 2.** (Continued)

|  | *n* or median [range] |
|---|---|
| Undergraduate Student | 5 |
| Diploma course student | 3 |
| Retail | 2 |
| Hospitality | 3 |
| Unemployed | 1 |

*Participant sexual identity was self-reported and listed here verbatim.

† Most participants were asked if they identified with any ethnic or religious group/s within their country of origin. Where an ethnic group was reported it is written here verbatim.

**Living as a gbMSM in country of origin**

1a. Law, religion, traditions and culture

1b. Fear of judgement, shaming and discrimination

1c. Repressed and hidden identities

1d. Exposure to HIV-related stigma

**Living as a gbMSM in Australia**

2a. Acceptance and freedom

2b. Sexual discovery and exploration

2c. Internalised fear of coming out

2d. Internalised fear around HIV testing

2e. Experiences of racial discrimination in Australia

Men described a transnational duality of lived experiences as a gbMSM in their country of origin and in Australia. Almost all men reported they felt the need to hide their sexual identity in their country of origin for fear of discrimination or judgement, and immigrated to Australia where they felt there would be more acceptance for LGBTQI communities. This duality presented itself differently among the participants in terms of varying degrees of discrimination they either experienced or were exposed to in their countries of origin and the level of acceptance they felt in Australia. The essence of their shared experience however, was that they had spent much of their lives hiding part of their identity and were now living in a place where they felt they had more sexual freedom. Despite changing environments, many participants described the effects of internalised stigma on their level of openness in Australia and on anxiety with HIV testing. Additionally, acceptance of their sexual identity in Australia was for many, coupled with experiences of racial and sexual discrimination racism.

## Living as a gbMSM in country of origin

### 1a. Laws, religion, tradition and culture

Men originated from 12 countries in Asia, a number of which have laws in place criminalising same-sex sexual activity (Pakistan, Malaysia, and Singapore). Of the 12 countries, only one had recently (2019) legalised same-sex marriage (Taiwan), however legislation was not passed until after the participants had migrated to Australia.

*Last year we had a vote about gay marriage but 70 per cent refused gay marriage in Taiwan.*

—Participant 10, Taiwan, 4 years in Australia

While several men were unsure of the legality of same-sex relationships in their country of origin, they felt that regardless of the law, being gay or bisexual was not acceptable in their societies.

*In China, I'm not sure if it's [homosexuality] criminalised or decriminalised, but I think we just don't care; the government. For the society—I'm not saying about the government—for the society, it's very hard to be gay.*

—Participant 13, China, 1.5 years in Australia

Men often spoke of firmly entrenched religious and cultural beliefs that strongly reinforced the acceptability of male and female unions only.

*. . .we know already that the Bible says man marries a woman so. . .I mean the way that we are thinking being religious people, I mean Roman Catholic, is against man to man. Very against it.*

—Participant 24, Philippines, 2 years in Australia

One participant noted that in Vietnam the unacceptability of homosexuality went so far as to not even recognise the concept in their language, the closest interpretation being to align it with paedophilia.

*[In Vietnam] basically the idea of homosexuality is considered as an idea that shouldn't exist, even as a concept. So, there's no word to describe homosexuality, it not even exists in the language. The only word that exists in the language, to describe homosexuality, is a word called pê-đê, which is basically a French origin word, and that mean paedophile, originally.*

—Participant 21, Vietnam, 3 years in Australia

For others, even if homosexuality was recognised, religious 'rules' and beliefs often dictated expected behaviours if one was LGBTQI, which largely centred around 'hiding' one's sexuality.

Cuz Malaysia's like a Muslim country. . . Like, there's certain things like you can't, you can't hold each other's hand, or you can't. . . There's certain rules, and, even if there's a gay club, or a gay bar or whatever, you can't even enter if you're Muslim. So they'll check your ID and everything.

—Participant 2, Malaysia, 1.5 years in Australia

While some participants felt these 'rules' were informed by religion, others attributed them to deeply embedded sociocultural traditions and conservative beliefs held in their society or region.

*The thing is, it's different back when I was in Indonesia, because I cannot stay open, because it's quite conservative there. I think that it's not only we have to hide [being gay], but the thing*

*is, we have to avoid, like, how people are easily persecuting [people because of] something bad, based on what they think is right.*

*—Participant 9, Indonesia, 1 year in Australia*

A few participants discussed differences in views of homosexuality in their local regions versus other cities in their countries of origins or the country of origin as a whole. One participant described the interplay of different communities' (religious, social, lifestyle) attitudes towards homosexuality in his small town in East Malaysia. To him, these differing community attitudes resulted in homosexuality being viewed as a sort of open secret—people living in the area know gay people are there and while they don't condone expressions of homosexuality, they also don't chastise them for their lifestyle.

*But in terms of, like, community wise, so for where I'm from, specifically the state of [state named], so that's in East Malaysia, so there's sort of this open secret, where it's, like, 'oh, that clique of people, they're openly gay', but no one will talk about them. There's no need to chastise them for their lifestyle, but we all know they are there, that kind of thing. I guess it's just because of–just certain–I don't know what it is, but I guess it's just like, sexual exploration is not really much of a thing back there, so yeah. . . And Malaysia being a Muslim majority country is that you just don't talk about homosexuality, as a general rule, but yeah, that's sort of the weird dynamic I was saying–like, yeah, but most–at least from my hometown, is Muslim. They've taken weird stances on homosexuality. . . like things like you can be gay but you don't have to express it. I'm not sure how to respond to that, so that's part of the restrictions within the community I was talking about. It's not even just the gay community, it can be the Muslim community, it can be within–because my hometown is a relatively small town, so I'd say people from the same high school, or the same colleges or uni[versity], they tend to know each other, so that's sort of a community within itself, yeah.*

*—Participant 11, Malaysia, 4 years in Australia*

Another participant spoke of his society's conservative values even restricting what clothes people can wear. To him, being openly gay in his country of origin was not fathomable since they cannot even accept men wearing shorts. He did not place the responsibility for these rules with his religion, but rather with the society and culture in his 'Old City', which he depicted as being more traditional than other cities in India.

I'm from the Old City. It's just that I don't blame my religion but then I blame the society and the culture. They are the ones that are setting the rules. The religion has never set the rules. . .I remember this one day I went out and my friend picked me up. I was lucky. I was like, "All right. I can just go quickly". I went down and sat in the car but then on my way back I was dropped somewhere a little bit far away, like a five-minute walk to my house. I cannot forget that five-minute walk, the names I was called and the looks I got were horrible . . . Forget being gay, I can't wear shorts. No straight man can wear shorts in my country. Not my country, just my particular society. It's so weird. Being gay would probably be the end of me.

*–Participant 6, India, 5 months in Australia*

## 1b. Fear of, judgement, shaming and discrimination

As a consequence of how LGBTQI persons were viewed in their society, most men described a feeling of oppression and fear around being judged or discriminated against in their country

of origin due to their sexual identity. Men often spoke of feeling different and alienated due to their society's judgement of people who are gay or bisexual.

> [It's the] attitude of the person and of our people. They are still saying, most of the people, that if you are gay or if you are bisexual or lesbian, you are the different people. It feels alien. . .Even though you are a Chinese, you are Asian, but you are not, cannot fit in any of the group of these people. So that's what I call judgmental.
>
> –Participant 17, China, 3 years in Australia

It was common for men to speak of their intense fear of others finding out their sexual identity, with some reporting they did not feel they would cope emotionally or psychologically with the judgment they might receive if others were to find out. A couple of participants went so far as to say they were afraid for their very lives if their sexual identity were to be revealed in their society.

> I don't know [why I am afraid]. Maybe because of people around me. Maybe the judgement. It might be affected emotionally, psychologically to me and I don't know how to adjust it. I don't know how to place myself in the public being a man but actually I do some stuff like this [have sex with men].
>
> —Participant 24, Philippines, 2 years in Australia

> Being gay, we're oppressed. We have to be in the closet. Our lives could be in danger.
>
> —Participant 8, Indonesia, 7 months in Australia

Others feared bullying and public shaming if they were to show any signs of affection in public with a same-sex partner.

> . . .in Singapore, if I were to hold my boyfriend's hand and walk down the street, people would be staring, people will take photos and posting them online. Singaporeans these days, they have like this mob mentality. . . like they are against these kind of things and just share kind of thing and people would see it. Like some people even got into trouble with their school and work because they were seen in public showing affection as a gay man.
>
> —Participant 16, Singapore, 3 months in Australia

A few men reported that in their country of origin same-sex activities were regarded as purely about sexual desire, devoid of real feeling or emotional connection.

> . . .if a man have sex with a man, it just purely about lust, about sexual feeling, there's nothing about true feeling, or anything like that. . .
>
> —Participant 21, Vietnam, 3 years in Australia

> . . .normally, these Asian people have this attitude about LGBTQI people are born to do sex. . .they are kind of sex workers, kind of prostitutes. . .
>
> —Participant 15, Sri Lanka, 2 years in Australia

## 1c. Repressed and hidden identities

As a consequence of entrenched religious and sociocultural views and beliefs around homosexuality in their countries of origin, and the fear of judgement and discrimination they felt

they would face for their sexual identity, men often reported hiding their identities to varying degrees. One participant described needing to hide not only his actions, but his thoughts from others to make sure they did not become aware of his sexual identity.

> . . .you always have to hide. If some other people know that you are gay they are always thinking bad about you and they will always–maybe talking bad words behind you, something like that. . .I have to be very discreet. I have to make sure that no one knows what I'm thinking, what I'm doing.
>
> —Participant 13, China, 1.5 years in Australia

Others spoke of concealing their sexual identities to make others think they were "normal" or heterosexual, which meant pretending to be a different person to how they truly saw themselves.

> Yes, even in my country I probably need to pretend so many things. Try to make people look at me as a normal person.
>
> –Participant 22, Laos, 1 year in Australia
>
> *. . . so whenever I'm around back home with my family or my friends, I kind of have to hide it or pretend to not be me.*
>
> –Participant 18, Laos, 4 years in Australia

For some men, feeling the need to conceal their sexual identity meant they had been unable to 'come out' (i.e. disclose their sexual identity) to any family or friends in their country of origin. Their fear of being judged, unaccepted and shamed meant they never intended to tell their families or friends about their sexual identity.

> *I didn't come out, actually, yeah. . . I think people there are still a little bit conservative, I think. . .especially in older people. . .Yeah, I think it's not a good idea, coming out in Taiwan.*
>
> —Participant 5, Taiwan, 3 years in Australia
>
> Back in Indonesia, so hard, especially to me, as a Muslim—you know, in Muslim rule, it's so hard to accept gay. It's so hard, my family is conservative religion, and that's why I'm never coming out with them.
>
> —Participant 14, Indonesia, 3 years in Australia

Some participants described being afraid their parents would disown them if they were to find out their sexual identity, while others were reluctant to share this information for fear it would cause their parents pain.

> The topic of disownment comes up a lot when I hang around with my friends. . .and I feel that as Asian queer people, we understand the feeling of, if my identity is out there I actually could lose my entire family. And I feel like that's an issue that not a lot of my friends will understand unless they're queer and Asian as well.
>
> —Participant 18, Laos, 4 years in Australia
>
> *I'm pretty sure my family would disown me even though they love me. . . My siblings do know that I'm gay. But my parents don't and I don't intend to come out to them because I don't*

*want to hurt them because they do have expectations for me even though I'm not fulfilling them. But I don't see the point of breaking that thing and hurting them. I know that I have to be myself but then it's okay. If I'm not fulfilling their dream, then I'm not supposed to at least break that dream. They can live in that bubble. I don't want to pop that.*

*—Participant 6, India, 5 months in Australia*

Only a handful of men reported they had been able to come out to members of their immediate family. One man described feeling some support from his family but at the same time feeling they were ashamed or fearful for him and wanted him to conceal his sexual identity from society.

*. . .but my immediate family knows that I am not straight and they support me. . ..But also, they tell me to keep it hidden because when we'd go out into society and meet other people in our circle, they don't want it to be a thing that I get labelled as. But I also feel like my parents don't want the shame that comes with that as well. . . .*

*—Participant 18, Laos, 4 years in Australia*

Some men reported they had come out to friends or siblings but not parents or older generation family members. Several men touched on there being more acceptance and understanding in the younger generation than the older generation.

I guess my family is from the older generation. So they don't really approve of it. But my friends are quite open, so yeah. I'm fine with telling them all about this.

—Participant 23, Singapore, 8 months in Australia

*So for young people they don't mind about gay in Laos, but for old people they quite mind yes.*

*—Participant 22, Laos, 1 year in Australia*

Some participants described having a supportive group of friends who were also gay in their countries of origin to whom they felt comfortable being open about their sexual identity. For these men, having gay friends gave them a sense of strength in numbers and the courage to be themselves in the face of judgement.

So back in high school my friends were—I surrounded myself with gay friends actually . . . even the teacher makes fun [of us]. They counsel us. Because it's a boys' school you have to be manly because it's stated in the [motto] of the school. . . it's a great school, I didn't complain. . . I remember going to the counsellor room and tell us 'why you want to do this, what would your parents think?'. . .[about us] being feminine . . .we try to bring a message [that] even though we are feminine, but we have discipline. We have the courage that you want to be. We show them all. So I'm kind of proud of that one.

–Participant 20, Malaysia, 3 years in Australia

Yes I would say I was a part of the gay community in Singapore because partly because Singapore is so small so like if you're gay you know who's who, that kind of thing. It's kind of nice because the population, the community is quite well knit. We have a lot of events and we always see the same few faces and just say hi and so on yes. . . Yes, that's a nice thing to see in Singapore. . .I guess like I got to know more gay people through those events, because

even though like, how do you say this, I've not made any gay friends through school or work or anything. . .I don't usually say out that I'm gay kind of thing back in Singapore. . .

–*Participant 16, Singapore, 3 months in Australia*

Other participants spoke about being open about their sexual identity to close friends in their countries of origins and not just those that were also gay. One participant explained how important it was for him to be open with his friends in China in order to feel like himself. For this man, having a boyfriend gave him the courage to be open to his friends about his sexual identity despite feeling that people may see him as abnormal.

*I got the courage because I have a boyfriend that time, so I got the courage to come out to my friends. . .The real reason is I want to come out because staying in the closet, it's a very hard life, I should say. You should pretend like you are like not a gay and they will think, 'oh, you're just like the other people who like girls'. But I don't want that. I want my friends [to] know truly about me. . . It's not normal [being gay], yes, to most of the people. They don't think it's normal things, but I think it's normal, we are. . . I think it's—after I coming out—it's a little, much easier. I don't pretend to be what these other normal people. . . I don't want to pretend to be someone. I just want them to [see] myself. Truly who I am is to be honest to my friends and to myself.*

—*Participant 17, China, 3 years in Australia*

## 1d. Exposure to HIV-related stigma

Entrenched in the religious and sociocultural views and beliefs around homosexuality in men's countries of origin, was HIV-related misinformation and stigma. Most participants disclosed being exposed to some degree of HIV-related stigma, which they predominantly attributed to the widely held stereotype or belief in their countries of origin that HIV was an illness that exclusively affects gay men.

. . .having HIV in the Philippines is already a–you know, it's something that you should be ashamed of. That's the perception. . .because gay people are being frowned upon, so yeah, and usually, HIV is associated with gay people.

—Participant 19, Philippines, 7 months in Australia

As a consequence, as one participant described, this led some men to keep their HIV diagnosis to themselves, as well as their sexual identity—in what he termed 'a very deep closet'.

They keep it [HIV diagnosis] to themselves, first because they're afraid that people would know that they're gay and secondly they are afraid that people will know that they have HIV. They have to live in a very deep closet and most of the time, in most of the cases, it's too late to ask for help. It's too late to ask for medication to make the virus undetectable. So it's pretty hard.

—Participant 8, Indonesia, 7 months in Australia

Several participants described a general view in society of HIV as being a punishment for the sexually deviant behaviour of gay men. In many of the men's countries of origin, the view was held that people who have HIV only have themselves to blame for contracting it, through their amoral actions which go against 'nature' or 'God's will'.

It means just having this awful disease which you got with your actions and this is the consequences you are facing now. You've done something wrong and God is punishing you. That is how it is seen because you have laid with a man and now you have to go with the consequences, exactly how it was in the eighties in America. It's just the same. We are still in the sixties, seventies, eighties in India.

—Participant 6, India, 5 months in Australia

Others explained that people living with HIV were considered unclean; a belief particularly prevalent again among the older generations.

*Like in my mum's, that generation or older generation, they think that it's a shame, it's really blame. . .This–shameful–it's shameful because they think that it's sexual disease. So it means you are a not a clean people. . .*

—*Participant 12, China, 1.5 years in Australia*

Many participants described a fear in their societies of getting physically and emotionally close to people living with HIV because of these beliefs and misconceptions around transmission. As one participant noted, in his country of origin a cancer diagnosis would elicit sympathy from others, whereas a HIV diagnosis elicits judgment, discrimination, assumptions around deviant and immoral behaviour and a fear of transmission if the person gets too close to them.

*Okay people look at people having HIV so badly, for example, you have cancer. . .people say, oh poor thing, he is unwell, he got cancer, I feel so sorry to him. People feel good to me if I get something else, but I'm not sure about okay people know he's having HIV don't stay close to him, you know you're going to get infection, no don't. Oh my God he probably have done something so bad, oh my God, they don't like it they hate it.*

—Participant 22, Laos, 1 year in Australia

Men often described how the HIV-related discrimination in their countries of origin resulted in an isolating and challenging life for people living with HIV, in terms of finding friendship, love and even a job.

*. . . there are a lot of serious discriminations on you so it's hard for you to live. It's hard for you to get a job; it's hard for you to make friends; it's hard for you to get a partner and get married.*

—*Participant 12, China, 1.5 years in Australia*

As a result of HIV-related stigma, participants often described a fear around HIV testing in their countries of origin. A couple of participants reported feeling judged or discriminated against while getting a HIV test, while others described the fear and anxiety they felt before testing due to the anticipation of being judged for being gay or of being an 'AIDS monster' and even a fear of being physically injured or killed for having the test.

The first time it took everything. I went to the local hospital, the government hospital and it was the anonymous thing as well. I mean, no identity and all but then I was scared because it was my city and if anybody got to know that I was getting testing for HIV, oh my God,

it's a big thing. . . these people will kill you just for getting tested [for HIV] because, "Why are you getting tested?" They'll look for that and, "Oh, you've had a sexual relationship with a man? You can't do this".

—Participant 6, India, 5 months in Australia

*My first time doing a HIV test was in Taiwan and that time I was very nervous and I wore a mask to do the test [to hide my face]. . . Because in Taiwan, there's some [perception that] if you are a gay man, you are just an AIDS monster. So the [perception is] if you are gay, you will contract HIV or AIDS.*

*—Participant 10, Taiwan, 4 years in Australia*

See Table 3 for further quotes describing sexual identity and HIV-related stigma in participant's countries of origins.

## Life as a gbMSM in Australia

### 2a. Acceptance and freedom

In contrast to their countries of origin where participants often spoke of having to hide their sexual identity, almost all participants described Australia as being more inclusive, accepting and open about people's sexuality. Many men described this acceptance in terms of feeling a sense of 'freedom' for the first time in their lives and the chance to finally be themselves and explore who they really are.

It's a huge difference compared to Singapore because here is like, everyone is so inclusive. They don't care if you're gay, they don't care whatever, race, religion, your beliefs whatsoever. It's like a huge difference, back in Singapore where everything is like, you have to hide

**Table 3. Men's quotes describing HIV and sexual identity-related stigma in their countries of origin and its impact.**

| | |
|---|---|
| *Discrimination in country of origin* | . . . it doesn't really matter the legal marriage, what really matters is the opinions, the views that people look at us. . .. You feel like discrimination, you feel like it is not comfortable.<br>—Participant 12, China, 1.5 years in Australia |
| *Hidden sexual identity* | Once they find out maybe they will judge me. It's just the judgement.<br>—Participant 24, Philippines, 2 years in Australia |
| *Fear of disownment* | I'm going to be kicked out from my family after I come out, so. . .No, [I haven't told them], not yet. I still want to call them, and I still want to contact them, and I want to know how they're going. . . so that's why I'm not telling the truth, but I really wanted to tell the truth, like I don't want to be lying all the time. My whole life was a lie, so yeah, I hate to be lying.<br>—Participant 15, Sri Lanka, 2 years in Australia |
| *HIV stigmatized in country of origin* | . . .for the general people, especially those from relatively smaller places like my home town, people's acceptance of HIV is very no. . .they have discrimination about HIV positive patients. Even though quite a lot of them are HIV positive because of some lack of education of protection ways. People think you get HIV because you are morally challenged.<br>—Participant 13, China, 1.5 years in Australia |
| *Judgement for getting an HIV test* | I wasn't too keen on getting myself checked, because even certain clinics, even the staff in the clinic, has expressed obvious dislike when you ask for a HIV check-up, so instead of asking things, like. . . ' have you done your check-up with us before', they'll go into 'what have you done?'<br>—Participant 11, Malaysia, 4 years in Australia |

yourself, you're kind of repressed. I feel like I can really be myself more here compared to when I was in Singapore.

–Participant 16, Singapore, 3 months in Australia

. . .being gay here, I feel embraced and I feel like I can be myself. I can just find everything that feels like 'that's me' so it's great.

–Participant 8, Indonesia, 7 months in Australia

When asked to describe life as a gbMSM in Australia, a number of men illustrated the freedom and acceptance they felt through examples of same-sex couples holding hands in public.

*People here are quite open. . . I've seen like lesbian and gay couples walking around the street holding hands and like, there was no, like, verbal abuse or anything towards them. So it's not that bad.*

—Participant 23, Singapore, 8 months in Australia

Well, at least here, I can hold hands with my partner in public, but we cannot do that in the Philippines.

—Participant 19, Philippines, 7 months in Australia

One participant described how emotional he felt when he first arrived in Australia and realised that he could show affection in public without fear of judgement or discrimination. To him, the freedom of expression he felt and continues to feel in Australia is one of the reasons he is considering staying.

*But just that freedom that I can wear whatever I want, I can hold hands with whoever I want and I can kiss whoever I want on the road. It's amazing and that's literally the best. I remember the very first day that I had here, a dude just told me, "You know what is the best thing about being in Australia? You can kiss on the road" and then he kissed me on the road and literally I had tears in my eyes. Just feeling that freedom was everything, yeah. It's honestly amazing and that's probably one of the reasons I might stay back in Australia.*

—Participant 6, India, 5 months in Australia

For others, the acceptance they felt in Australia was tied to the legalisation of same-sex marriage.

*[Australia is different than Indonesia] because, luckily Australia, the law gay marriage already passed, that's why so happy. . .*

—Participant 14, Indonesia, 3 years in Australia

## 2b. Sexual discovery and exploration

For many participants, the freedom and acceptance they felt in Australia to discover themselves for the first time meant they felt able to talk about and explore their sexuality both physically and emotionally.

*Relatively to Malaysia, it's been very open, so that means in terms of being, physically, like I'm more comfortable and more relaxed with talking about certain things with colleagues, friends,*

*but also in terms of sexually, as well, it's more open to exploring here, whereas I would say comparatively, back home, there is a certain degree to things that we're allowed to say or do.*

—Participant 11, Malaysia, 4 years in Australia

For some participants, the independence from their families and social connections in their countries of origin were the main facilitators to sexual exploration in Australia. They felt the freedom to explore their sexuality without fear of their sexual identity being exposed.

*As a bisexual, since I'm the only one here so I can do whatever I want. I actually try to experience the things that I wasn't able to experience back in the Philippines because I was living with my parents. So I'm trying to discover a bit—my discovery there is limited compared to here on how to make myself more satisfied or whatever I want to do.*

—Participant 24, Philippines, 2 years in Australia

One participant attributed the freedom he felt to explore his sexuality to the safe 'sex positive' environment he feels in Australia. For him, the normalisation of expressing your sexual desires and boundaries made him feel like he was safe to explore.

*A lot of things that I observe, first of all how people are just openly sexual. The term that I heard was sex positive. If you need to talk about sex, it's just easy to engage and hook up. I like the culture of like, straightforward, you know, you say what you want, say [what] you're not comfortable [with]. For me, it's more comforting in a way. I have [a] secure space and place and I can just explore everything because I don't need to be afraid about myself and my security.*

—Participant 8, Indonesia, 7 months in Australia

One man described the freedom to explore his sexuality in tangible ways, such as the ease of buying condoms without judgement and access to smartphone dating applications without restrictions.

*. . .just like having a lot of people who have the same identity as I am, it's quite easy for me to have the access of having sex stuff, because personally, I think that it's easier for me to buy condoms here, and then after that, there's no limitation on–for example, it's easy for me to find guys, because the dating apps is actually not blocked here. Back there, in Indonesia, it was blocked, so the thing is, it's not restricted.*

—Participant 9, Indonesia, 1 year in Australia

### 2c. Internalised fear of coming out

Given the level of acceptance felt in Australia compared to their country of origin, most men described feeling they could be more open with their sexuality in Australia. However, many participants described some hesitation or limitation with how open they were willing to be with their sexuality, preferring instead to keep their sexual identity to themselves unless asked directly or only disclosing to people they felt confident would accept them.

For these men, the internalised and residual fear of coming out as a result of their experience in their country of origin reverberated to their lives in Australia. The stigma, judgement and discrimination they were exposed to or experienced in their country of origin, was still strongly ingrained and difficult to dispel despite moving to a more accepting environment.

*. . .moving from one environment to another, it doesn't automatically change everything. You carry with you sort of behaviour from the old community, to the new one, so it's actually take me a while to actually untangle all that, but now, I have a gay group of friends, I'm pretty open to my housemate, and a lot of people know that I'm gay, so in a way, I think I'm more open now. At the same time, I'm still very reserved. . .*

—Participant 21, Vietnam, 3 years in Australia

*I would say, I guess it's just something that's been reinforced over the years, like growing up. . .back home, if anyone asks me, I would try to beat around the bush a bit. . . but here– even though I'm comfortable, but I guess that mindset is sort of just ingrained, 'let's not share that too much', yeah.*

—Participant 11, Malaysia, 4 years in Australia

One participant shared how he found it immensely difficult to be open about his sexual identity for the first time. He chose to come out to a close friend he had known in Australia for one year but he described experiencing great trepidation, fear and anxiety around telling his friend. For this man, hiding his sexual identity for so long was a huge burden on him, and he described great relief when he came out and was accepted by his close friend, for the first time in his life.

*I was so scared that time, when I'm telling to him, I was so scared, and he pushed me, just tell me. So, I thought about it again and again, should I tell this, should I tell this? Then, finally I tell, like, and then he told, alright, then he told that's nature, and science, and he hugged me and told it's alright to be my friend. I told, like, I really don't want to lose my friend, but then he told, 'no, you're never going to do that, I'm never going to do like that, I will stay'. . . gave me that kind of feeling like, alright, I'm alright to live in here, in this world.*

—Participant 15, Sri Lanka, 2 years in Australia

Others reported feeling they could come out to some friends in Australia but not others. For some this related to friends having connections to their country of origin and a fear of their family finding out through those connections.

*The thing is, I cannot [come] out to everyone, just because I still have some Indonesian community here–I mean, like, I have some relatives to those conservative Indonesians, here. . .but for others–for example, my community in the uni[versity], and then some key friends, I came out to them.*

—Participant 9, Indonesia, 1 year in Australia

For others, not being able to disclose or come out to friends in Australia who were from their country of origin related to concern they might lose those friendships for fear their friends would not be accepting of them, having grown up in environments not as accepting of LGBTQI communities as Australia.

*I came out few friends, not any Sri Lankan friend yet. I have one [Sri Lankan] friend, really good one, but I don't want to come out in case—you know, like. . . I really don't want to miss someone, because I never had a friend in my life, so if I got a friend, I'm trying to pro- tect him, and I don't want to lost my friend. . .he will not [accept me]. . .one day he told me,*

'how do I deal with these gay people, I hate these' something [like that]. I'm not going to tell him.

—Participant 15, Sri Lanka, 2 years in Australia

*If they were born in Indonesia, I don't come out. If they were born here, I coming out. I just, you know, ask about gay–how [do you feel] about gay? I just ask their opinion. If their opinion like, 'oh, not a gay, blah blah blah', I'm not come out to them.*

—Participant 14, Indonesia, 3 years in Australia

## 2d. Ingrained fear around HIV testing

Many participants described neutral or even positive experiences when they had had a HIV test in Australia, particularly contrasted with their countries of origin. However, many described residual or ingrained fear of HIV testing in Australia as a result of the sociocultural environment in which they grew up, where HIV was heavily stigmatised.

Several men reported they still felt anxiety around HIV testing in Australia, which largely centred around the perceived negative impact a HIV diagnosis would have on their lives.

*. . .it's [getting tested for HIV] quite a bad experience. On that day, I feel nervous all the time, I feel nervous, I feel scared. I feel that if I get HIV, how do I stay in this world, it's like in my brain, I keep thinking all the time.*

—Participant 4, Thailand, 4 years in Australia

A few participants even described delaying testing due to fear of finding out they were HIV positive and what it would mean for their lives. One participant, who had a close circle of friends in his country of origin described how difficult it would be to open up to them about an HIV diagnosis, let alone to his family to whom he was not comfortable being open with his sexual identity. To him, delaying an HIV test was a result of the anxiety he had about what a positive result would mean for his life.

*Yes I think anxiety might have played a part [in delaying HIV testing] as well. . .Like I don't know, I guess it's better not to find out the bad news if you never get yourself to know; I guess that's why. . . I'm not sure if I can ever tell my family about it [an HIV diagnosis]. . .They would want to know how I contracted HIV and I'm like not even out to them. . .Yeah so I don't think I could ever bring myself to tell my parents especially. I guess my social circle would be quite understanding but I don't think I would be that courageous to tell them 'oh I have HIV'. I would have to like build up courage slowly and tell them about my situation now. It would definitely put my, how do you say, my focus and my courage levels would go down. . . Right now I still have struggles with anxiety like day to day life and all and this would really put like a huge impact. I would be a total mess I would reckon, yes.*

—Participant 16, Singapore, 3months in Australia

One participant had his first experience of chemsex (i.e. use of recreational drugs during or before sex) in Australia during which he had sex without a condom and described the fear and anxiety he felt about getting an HIV test in the following weeks. He described feeling like he was HIV positive after the night of chemsex but was too scared to get tested because of what an HIV diagnosis would mean for his life, particularly the isolation he feared he would face living with HIV in his country of origin.

When the whole drug thing happened with me, the whole five weeks, every single day I was thinking, 'When do I go? When do I go? Why am I not going? What are you doing with yourself? Why not? Why not?' I actually thought I was positive. . ..[Having HIV] it would impact in a lot of ways. . . Now that I'm positive, who's going to be with me? . . .[HIV is seen as] Untouchable. . . Not a lot of people would know about it. Only a close set of people would know about it. I don't even know if I would tell my parents if I got positive because they'd just break down and they would be–I don't know. I've thought about it a lot of times. I don't have it but you never know.

—*Participant 6, India, 5 months in Australia*

Others delayed HIV testing for fear of the test itself and worrying about how clinicians would treat them given their previous negative experiences in their countries of origin.

*I just didn't know how people would treat me [if I came for a HIV test], especially at the beginning. When I was in [college], everyone around me was, like, mostly Chinese students, and I didn't really deal with Australian people, and I don't know how they would see me. I don't know their opinion on anything of me. I just kind of feel like they would be similar to Chinese people. . . I didn't feel safe to do that [get tested], yeah.*

—*Participant 7, China, 1.5 years in Australia*

For many others however, the experience of HIV testing in Australia was positive and had become easier and less anxiety provoking due to feeling supported and unjudged by clinicians.

*It took me courage to come here [for a HIV test] as well because I had no idea how it worked and everything. But then it was pretty easy the moment I walked in here. It was really easy. . . It's good. It's just quick and fast. People care here. I feel that. I mean, if I was getting tested back home, nobody would ask me this many questions and offer counselling.*

—*Participant 6, India, 5 months in Australia*

*But more and more, I feel like, especially doing it here, I feel more relaxed because everyone feels, you know, whatever the results, we've got your back, you know, free medication if you feel like you want to have it, "We're going to support you" and stuff like that. I feel more secure. So I feel like nowadays it's just obligatory and the anxiety is not as big as my first time or my second time. But still, of course there is an anxiety.*

—*Participant 8, Indonesia, 7 months in Australia*

## 2e. Experiences of racial discrimination in Australia

While men reported a much greater sense of freedom and acceptance around their sexual identity in Australia, and an opportunity to finally explore their sexuality and who they were, they also described the challenges of being a part of an ethnic minority group in Australia. Most men described experiences of discrimination and racism in Australia, particularly in terms of sexual racism, but others also felt they were subject to racism due to their English-speaking abilities. A few men framed their experience as a sacrifice; a trade-off between being sexually free in Australia and 'racially open' in their countries of origin. For some participants, these feelings of discrimination extended to the 'white-dominated' gay community in Australia where they felt they did not really fit in.

Smartphone dating applications appeared to be the predominate source of sexual racism, with men reporting seeing phrases like "No Asians" on profiles of men on dating apps. Most men describing sexual racism on dating apps were quick to follow up their observations with statements about how it did not affect them or how they try not to judge the men making racist comments. Despite raising these examples, some men felt unsure if these experiences were actual racism, or just expressions of personal preference.

> When I was noting on Grindr [a smartphone dating application for gbMSM]. . .some people put in their profile or description, they might be saying, putting in words like 'Caucasians Preferred'; 'Asians Preferred' or even some words; 'No Asians'; 'No Indians', that happens. But I guess everybody has a preference. So usually when I see such words I don't think so much about that. I'm not sure if that is racist

> —Participant 13, China, 1.5 years in Australia

> So, dating apps is, I'd say, one of the ways that I'm more comfortable with meeting other men, so I guess that's where I experience it [racism] the most. . .. I've sort of just accepted it, to be just sexual preferences, but yeah, of course there are moments where I just think, well, maybe it's not because of preferences, maybe they're just not into Asians, maybe. Yeah, so I can't decide if that's–they always mask it as saying it's just a preference. So, not sure where I stand on that. It's, like, well you're right, it is preference, but at the same time, it's like, hmm–it is a bit racist, isn't it?

> —Participant 11, Malaysia, 4 years in Australia

For other men, the discrimination and perceived racism they described occurred in public venues or places; though yet again, men commonly reported they did not take the experiences personally or rationalised that the person they perceived as being racist was maybe just having a bad day.

> I wouldn't say overt racism, not in a way where I've been physically harassed, but I think there has been a few times when I've been on a tram, minding my own business. And then, because I'm standing in someone's way just a little bit, there was this Australian woman once who came up, I'm sure she's from the North of Australia, and she was like, go back to China. I was like, yep. . . that's okay because she's probably having a bad day and I don't take that personally.

> —Participant 18, Laos, 4 years in Australia

A few participants noted that the racism they have felt in Australia usually stems from their not speaking English as fluently as Australians.

> *I think that racism is actually not related to gay stuff, but it's more into the idea that we are actually not from Australia, and the way how we communicate is not as fluent or as fluid as what Australian capable to do.*

> —*Participant 9, Indonesia, 1 year in Australia*

Others felt it came from a lack of cultural understanding.

> *I experienced that [racism], but for me, I think that is actually not the reason why I have to blame some people, or somebody, just because of their incapability of accepting my ability to*

*speak [English]. I know that we're different, and I know those people are actually having limited understanding on that. . . I just see that they are actually not really educated on how to be culturally aware to people from different backgrounds, or be more accepting to people who have different forms of English. . . I don't say that that's insulting, but I can say that it's annoying.*

*—Participant 9, Indonesia, 1 year in Australia*

Some participants spoke of the trade-off of moving to a country where they are considered a racial minority, in order to have acceptance of their sexual identity. For these participants, feeling like they were more accepted for their sexual identity in Australia came at the expense of living in a racial minority group, and the challenges of discrimination that come with that identifier.

*Then I feel like no-one wants to approach me just because I'm Asian in a place where there's mostly white people. Back at home, of course I don't feel that. . .I didn't realise it up until I talked about it, in a way. So over here, I can be sexually open but cannot be racially open.*

*—Participant 8, Indonesia, 7 months in Australia*

*There is a lot of freedom here, and you can be gay. . . back in Pakistan, you cannot be openly gay. . . but on the other hand, there is a lot of racism, so it is good and bad both. . . it is difficult sometimes, really, really difficult, but then, you have to put up with it [racism] because the gay life here is much more better than Pakistan. There is no gay life there.*

*—Participant 1, Pakistan, 2.5 years in Australia*

Some men also reported feeling excluded from the 'white dominated' gay community in Australia due to their race. They felt there was only room for Asian gay men to be accepted into the community if they fell into specific categories, like funny or wealthy.

*It's very difficult for me to get into the white dominated gay circle. Having this Caucasian boyfriend helped a little bit, but still you can feel the differences are still there. . . the white Caucasian gay community, they prefer Asian gay guys that are more like funny guy, not someone who's very serious like me. Yeah funny is the first requirement for them to get into the circle. They can regard you as a friend but it's just a friend, it has nothing to do with real core circle of the community. You can't really get into that part. . .*

*—Participant 3, China, 4 years in Australia*

*I think it's because sometimes to fit in gay society, you have to be. . .you have to have an amazing lifestyle, amazing Instagram, this and that and be out there and do things and have money. I don't have money. I can't do all this, have an amazing life and have an amazing Instagram, buy fancy clothes, go to parties and stuff. I can't do that and honestly, I think to be a part of that society, you have to have all these things.*

*—Participant 6, India, 5 months in Australia*

One participant felt that part of the difficulty in feeling a sense of belonging in the gay community in Australia was a lack of representation of Asian LGBTQI people in the media.

*. . .even with the media, you don't actually see a lot of Asian images in the way that people portray queer identity. It tends to be–like. . .two white, gay men, and one is very queen-like, or something like that.*

—*Participant 21, Vietnam, 3 years in Australia*

Another acknowledged that despite feeling excluded from the gay community in some respects, there was a section of the community he felt welcomed into and felt a sense of belonging with these people.

But then again, there is a different group of people who are accepting and welcoming. I would say there is a ratio, like 70 per cent of people are like this but then the rest, 30 per cent of people–I do feel like I belong here, like I belong between these people. When I go to the voluntary things and stuff, the LGBTQ, the people just see the heart, not the outside but they see the inside.

—*Participant 6, India, 5 months in Australia*

While most men reported experiences or feelings of sexual and racial discrimination in Australia, not all did. A couple of participants described feeling like they were accepted and not judged.

*To be honest I don't feel any racism. Since I came here, cuz, I will say especially in the city, there's, half of it is Asian as well and you don't feel that you're like, you're Asian, and you're like uh, being separated from, yeah like they don't judge you where you come from.*

—*Participant 2, Malaysia, 1.5 years in Australia*

See Table 4 for further quotes on men's experiences of living as a gay or bisexual man in Australia.

## Discussion

In this study that describes the experience of newly-arrived Asian-born gbMSM—a group recently emerged as being at increased risk of HIV in Australia—we found that most described hiding their sexual identities in their country of origin because of fear of stigma and discrimination related to sexual identity and HIV. These experiences translated into significant anxiety about being tested for HIV in their country of origin as well as in Australia and resulted in delayed testing. Internalised fear of stigma and discrimination is likely to be an obstacle to achieving timely and regular HIV testing in this population.

Participants attributed the fear, judgement and discrimination with regards to their sexual identities in their countries of origin to the differing traditions, religions and cultural values

**Table 4. Men's quotes on their experience of living as a gay or bisexual man in Australia.**

| | |
|---|---|
| *Acceptance of sexual identity* | . . .when I came to here, I felt that freedom, independent, like you know, the people are accepting me as a human being.<br>–Participant 15, Sri Lanka, 2 years in Australia |
| *Hesitation to be open with sexual identity* | . . .if they ask I wouldn't deny it. . . but I wouldn't go out telling around like, oh I'm gay, I'm gay, that kind of thing.<br>—Participant 16, Singapore, 3 months in Australia |
| *Anxiety around getting tested for HIV* | I was kind of reluctant coming here [to the sexual health clinic], because I don't know how people would see me.<br>—Participant 7, China, 1.5 years in Australia |
| *Sexual racism* | if you go onto the apps, there will be guys that just–you know, like, expressively say that, 'no Asian', 'if you're Asian, don't talk to me', something like that.<br>—Participant 21, Vietnam, 3 years in Australia |

that shape their societies' view of homosexuality. As a result of these conservative views, most participants described hiding their sexual identities to a large degree in their countries of origin, a finding reflected in the literature from several countries in Asia [18–20]. While men often spoke of hiding their sexual identity to family members, it is important to note that there were some that were open to friends in their countries of origins, including a couple of men who discussed the strength they found from having friends in their countries of origin who were also gay. While measuring resilience (i.e. the ability to recover after experiencing adversity) among our participants was outside the scope of this study, peer support is a known protective process that leads to the development of resilience [31,32], which in turn may have a positive effect on mental health and well-being [33]. In our separate paper from this study (as yet unpublished), we discuss our findings that social support was a facilitator to sexual health within this group, notably some participants' positive experiences with local LGBTQI organisations that offer peer-led social support workshops. Previous research has also found that social support can impact sexual behaviours [34,35]. Increasing awareness of and opportunities to engage with peer-led services may be beneficial for men in this group in terms of bolstering their social support in Australia. Additionally, it is important for these services to have an awareness of the stigma and discrimination Asian born gbMSM can experience in Australia to enable them to factor this into culturally appropriate services which meet their needs. Further research could be done to examine mental health, resilience, and social support within this group and the implications for HIV prevention.

Additionally, participants described several forms of HIV-related stigma in their countries of origin: from viewing people living with HIV as unclean or shameful to fearing discrimination or even death for getting an HIV test. This finding is reflected in the literature from China, which shows high prevalence of HIV-related [36] and sexual identity-related stigma [18] among gbMSM not living with HIV. Many men in this study expressed continued anxiety about getting tested for HIV after arriving in Australia due to the negative experiences and HIV-related stigma pervasive in their countries of origin and for some men this resulted in delayed HIV testing. Because testing is a key component of HIV prevention programs, fear of testing is likely to lead to reduced participation in HIV prevention.

While almost all participants described feeling acceptance and sexual freedom in Australia, some were still hesitant to be open with their sexual identity due to deeply embedded and internalised feelings of shame and stigma around being gay or bisexual. For men living in countries with structural stigma towards sexual minorities (for example laws criminalizing sexual activity or discriminatory cultural attitudes) sexual identity concealment serves to reduce discrimination and victimization [13]. Our finding that some men continue to conceal their sexual identity even after moving to a society where they feel they do not face this stigma suggests this stigma becomes enmeshed and has an enduring affect, the repercussions of which could be investigated in future studies.

Despite feeling acceptance of sexual identity, some participants described racial and language fluency-based discrimination in Australia in several ways; from experiencing perceived racism because of their English language skills, to widespread sexual racism on dating applications. Some men felt unwelcome in the gay community in Australia as an Asian-born gbMSM; a worrying finding given the importance of social support on mitigating the negative effects of stigma [37,38]. Our findings reflect those in a similar qualitative study from 2018 of Chinese and South Asian gbMSM living in Auckland, New Zealand; this study indicated that men in this group were hesitant to share their sexual identity with others, including healthcare professionals, despite feeling personally comfortable with identifying as gay or bisexual [39]. That study also indicated that due to the racial discrimination they faced they had weak connections with other gay and bisexual men, and thus were in a potentially vulnerable position [39].

However, it is important to highlight some of the participants' responses to racial discrimination in Australia, notably how many explained that they do not take sexual racism on dating apps personally, but rather see it as men having preferences. Similarly, several men when describing instances of discrimination, saw the experience as an indictment of the perpetrator of the discrimination and not themselves, for example rationalising that the perpetrator was having a bad day or lacked cultural understanding. These responses to racial discrimination indicate a level of self-compassion among the participants [40]. Previous research among students in the USA suggested that self-compassion may play an important role in coping with stigma among those students who were in both a sexual and racial minority [41]. Future research could investigate the level of self-compassion within this population and its effects on resilience in the face of discrimination.

This study highlights the complex set of factors that contribute to HIV-related vulnerability in newly-arrived Asian-born gbMSM. Previous reports have shown that minority stress is an important contributor to psychological distress among gbMSM [22] and that sexual identity concealment is a critical mechanism linking sexual minority stigma and depressive symptoms [42]. The findings from our study suggest the potential for considerable minority stress among Asian-born gbMSM newly arrived in Australia, not only in terms of sexual minority status but also in terms of race and ethnicity in Australia. Central to this intersectionality is the extensive HIV-related and sexual identity-related stigma faced by this population.

There were several limitations to this study. First, men were recruited from a sexual health centre and thus the experiences of these men reflect views from men already connected in some capacity to sexual health services. Similarly, men needed to have sufficient English language proficiency to conduct the interview. Thus, men who are not connected to sexual health services or those with limited English have not had their voices heard in this study. It is possible that members of these groups may be even more vulnerable to HIV due to their further marginalised status, as not speaking the host country's language has been a reported barrier to HIV testing for migrants in high-income countries [43]. Additionally, these men were willing to discuss topics with the researcher that they felt were taboo in their countries of birth and therefore they may be more open than other newly-arrived Asian-born gbMSM. This study did not ask participants to describe the degree to which they identified with a particular ethnicity, but recent research has questioned the role of ethnic identity as a protective buffer for the stress of discrimination [44]. Future research could investigate whether men in this group with stronger ethnic identity have ameliorated stress levels in the face of discrimination.

An important limitation to this study is that the experiences described by the participants in this study were centered around their sexual identity and attitudes towards HIV, and thus their responses are not indicative of their entire lived experience in their countries of origin, nor their holistic attitudes towards living in their countries of origin. The primary aim of the study was to explore strategies that newly-arrived to Australia Asian born gbMSM were using or preferred to use to prevent HIV infection, in light of rising rates of HIV in this population. This study was therefore designed to explore participant's knowledge levels about HIV prevention strategies, and what may have shaped their knowledge levels. In asking men about their knowledge levels we asked men what it was like to live as a gbMSM in Australia compared to their home country whereby they commonly described differences in acceptability of their sexual identity in their countries of birth versus in Australia. Given the topics that were explored and arose, the results of our study may lend themselves to more negative aspects or experiences of their countries of origins". Similarly, while we asked participants if they identified with a particular ethnicity or religion, we did not specifically ask about perceived differences between ethnicities or religions in their countries of birth with regards to attitudes toward sexual identity and HIV. Further research is warranted to better understand how ethnic identity and

religion influences attitudes toward sexual identity and HIV among subcultures in different Asian countries.

Further to this limitation, "Asian-born" is an umbrella term applicable to a vast array of diverse ethnicities, societies and cultures, a fraction of which were discussed here. With regards to the topics presented here, there are, in addition to the variations between countries, variations in religions, cultures and attitudes within countries, as our participants touched on. Our aim was not to provide a generalizable finding applicable to all gay or bisexual men born in Asia and living in different geographic locations throughout Australia, but rather to capture the depth and breadth of experiences from people in what has been identified as a HIV vulnerable population in order to further direct targeted HIV prevention in this group. Given the sparsity of research into Asian-born gbMSM migrants living in Australia, the diversity of our study may in fact be considered a strength as our sample included men from a broad range of Asian countries, of varying ages and education levels. By focussing on newly arrived gbMSM we have been able to include narratives from several countries that have differing family, religious and cultural influences.

Another limitation that should be considered is that the interviews were all conducted by one researcher (TRP), a white female. Participants may have been reluctant sharing information with someone they did not feel was a peer or who could not fully understand their lived experience as an Asian born gbMSM. Further to this end, TRP and JEB (another white female) were responsible for the majority of the data analysis, with TRP doing the crux of the analysis and JEB conducting cross-checking on a sub-set of transcripts to confirm the coding framework and thematic analysis. One of the risks with qualitative research is that researchers can unintentionally bias various aspects of a study, with their own beliefs, values or preconceptions, which can result in an 'outsiders' view or interpretation of the data [45]. Despite TRP and JEB meeting regularly with the wider research team to discuss findings and challenge their assumptions about the attitudes and cultural backgrounds of the participants, it is possible the data were interpreted through their own cultural and gender lens as white female researchers. Additionally, had two to three researchers coded the entirety of the data (cross-coding/multiple coding) instead of one researcher doing the majority of the analysis, this would have provided more rigor to the analysis. While it would have been ideal to undertake cross coding of the data using multiple coders, which would have further assisted in mitigating bias, this was not possible due to time and resource limitations. This approach serves as a valuable strategy where these limitations apply [46].

A further limitation of this study was that participants were only offered a relatively narrow form of member checking, whereby they were asked if they would like to check the accuracy of their manuscript. While all participants indicated they were happy to be contacted again following the interview, none took up the offer to check their manuscripts for accuracy. In hindsight, having a broader approach to member checking, whereby men were re-engaged at the point of data interpretation to participate in a member check interview to offer their views on interpretation or their own interpretations of the data, would likely have ensured a more rigorous and potentially more accurate understanding of the data, thereby minimising the risk of a 'outsider' conflation or construction of Asian culture and Asian gay men's experiences. If broader member checking activities had been employed, promoting a more inclusive approach to the research, men would likely have re-engaged at a greater level, allowing for a more nuanced understanding of their experience.

This study contributes to the growing body of knowledge on the challenges facing Asian-born gbMSM living in Australia as well as enhancing the research surrounding HIV and sexual identity-related stigma. Our findings are consistent with previous literature showing prevalent sexual identity stigma in several countries in Asia and the impact of stigma on sexual identity

concealment. Additionally, our data highlights the potential discrimination Asian-born gbMSM face in Australia, which has implications for social connectedness, particularly with regard to LGBTQI communities and HIV testing practices. Future studies should determine effective strategies to reduce sexual identity and HIV-related stigma in newly-arrived Asian-born gbMSM.

## Acknowledgments

We would like to thank the study participants for sharing their experiences and time with us.

## Author Contributions

**Conceptualization:** Tiffany R. Phillips, Nicholas Medland, Eric P. F. Chow, Christopher K. Fairley, Jason J. Ong.

**Data curation:** Tiffany R. Phillips.

**Formal analysis:** Tiffany R. Phillips, Jade E. Bilardi.

**Investigation:** Tiffany R. Phillips.

**Methodology:** Tiffany R. Phillips, Nicholas Medland, Christopher K. Fairley, Jason J. Ong, Jade E. Bilardi.

**Project administration:** Tiffany R. Phillips, Kate Maddaford, Rebecca Wigan, Jason J. Ong.

**Supervision:** Jade E. Bilardi.

**Writing – original draft:** Tiffany R. Phillips.

**Writing – review & editing:** Tiffany R. Phillips, Nicholas Medland, Eric P. F. Chow, Kate Maddaford, Rebecca Wigan, Christopher K. Fairley, Jason J. Ong, Jade E. Bilardi.

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
