## [Decision Letter · Decision Letter 0]

15 Jul 2020

PONE-D-20-17766

’Moving from one environment to another, it doesn’t automatically change everything.’ Exploring the transnational experience of Asian-born gay and bisexual men who have sex with men newly arrived in Australia

PLOS ONE

Dear Dr. Phillips,

Thank you for submitting your manuscript to PLOS ONE. After careful consideration, we feel that it has merit but does not fully meet PLOS ONE’s publication criteria as it currently stands. Therefore, we invite you to submit a revised version of the manuscript that addresses the points raised during the review process.

Clearly the reviewers' individual recommendations vary; however, what is common to both reviewers, not incidentally both of whom occupy subject positions perhaps more aligned with the participants than the two apparent main authors who conducted the interviews and data analysis--not necessarily a critique, but a relevant observation--is a strong questioning of several presumptions that underlie the text and the analysis. These coalesce in the apparent homogenization of what are vastly different Asian cultures, ethnicities, religions and participants, and what emerges as a characterization of these men's 'home' cultures as unilaterally stigmatizing, homophobic, and seemingly antithetical to gay men's mental health. On the other hand, the description of methods and some of the analysis appears very thoughtful, supported by rich data, and guided by theory--all of which are strengths of the manuscript. And both reviewers indicate the import of the subject and strong potential for meaningful contribution to the literature.

I would invite the authors to seriously engage with the reviewers' comments. On the one hand, as above, I do think they raise meritorious concerns, for which I will pose the following suggestions:

1) When you discuss reflexivity, please consider  the subject positions of the apparent lone itnerviewer and data analysis persons; how might these impact on what the participants reveal in the interviews; what questions are asked; how the data are analyzed; conceptualizations of their 'home' countries; and the potential for a somewhat monolithic treatment of the construct of "Asia" and "Asian gay men"? This is in no way to say that only same-ethnicity, same-gender, etc. researchers can do valid research with like participants; but as the authors seem to be aware, and particularly in constructivist qualitative approaches, their identities do matter; and these need to be addressed in more depth. It is also generally a weakness in qualitative inductive approaches to have data analysis almost entirely conducted by 1 person--please note this as an additional limiation;

2) Are there no strengths revealed that emerge from an of the 'home' cultures? Any redeeming value of the experience of growing up as a gay man in these countries? I would encourage the authors to critically assess how their portrayal of Asia, while unintentional, may be read by others, in a rather flat, uniform, and quite negative context for all gay men; 

3) What of the strengths and resilience of the participants? Arguably the weight of the narrative construction presented here is that of 'victims' of deeply internalized stigma from their home countries and cultures, and additional stigma received in Australia. For ex., how do some or most Asian gay men immigrants to Australia *not *contract HIV? Qualitative research also can use 'negative case examples' as a means to shed more insight and more nuances in the data so as to avoid universalistic constructions, particulalry of 'victims'. This would help to provide a more full picture of participants as also possesing agency and strength; 

4) What might it say that all participants "declined" member checking? While one cannot lay 'blame' on the researchers, it is at least worthwhile to conjecture about what might have created a more welcoming or inclusive approach to the research that at least a few participants would indicate willingness to re-engage? It also seems as if the authors adopt a very narrow view of member checking. Were participants invited to offer their own interpretations of the data? or only to review the apparent accuracy of the transcripts? This is quite important as a successful approach to member checking might have helped to 'correct' what both reviewers read as a rather 'outsider' conflation and construction of "Asian culture" and "Asian gay men". This is also a limitation of the research that needs to be identified;

5) Importantly, you must provide a full citation for the other article noted as one in which some of the data have already been published; this is imperative both to assure reviewers and readers (and editors) that the articles are not redundant, and also for you to briefly but clearly indicate how the present manuscript is different from the other manuscript mentioned based on the same study;

6) As reviewer 2 notes, and I concur, it would be helpful to identify potential implications for HIV testing education, outreach, services, providers in Melbourne. It might be read as if the the full responsibility is on these individuals to navigate what may HIV services as fixed entitites. How might such services recognize the stigma and discrmination within the Australian context and create more culturally appropriate services to meet participants needs and encourage earlier HIV testing?

7) I would suggest that while it is plausible that the sample in recruiting from sexual health clinics identified the most 'willing' of participants to engage with services, such that other Asian gay men may be every more 'vulnerable', is it not also plausible that some Asian gay men who are less 'integrated' into Australian institutions possess cultural capital and cultural strengths that are protective factors against stigma and discrmination? Indeed this is found in a good deal of research on immigrants in many majority cultures.

8) The authors admirably note use of the COREQ guidelines. It would strengthen the submission to attach the COREQ checklist and indicate where in the manuscript each of the guidelines were addressed. Not every manuscript satisfies all of the items delineated, but evidence to show how the authors met many of these guidlines would be very helpful. This also relates to comment 1, which requires a fuller and perhaps more earnest engagement with reflexivity. For ex., see: Tufford, L., & Newman, P. (2012). Bracketing in Qualitative Research. *Qualitative Social Work*, *11*(1), 80–96. https://doi.org/10.1177/1473325010368316

Finally, while the authors need to address the core concerns shared across the reviewers, and as described above, I do not necessarily expect that each and every comment of the 2nd reviewer needs to be acceded to. However, if there is some misinterpretation by the reviewer in certain cases, I would ask the authors to briefly acknowledge and explain it. In other cases, the revisions as suggested should be incorporated. 

We look forward to receiving your revised manuscript.

Kind regards,

Peter A Newman, Ph.D

Academic Editor

PLOS ONE

Journal Requirements:

2. Thank you for including your competing interests statement; "NM, EPFC and JJO have received a research grant from Gilead Pharmaceuticals to conduct this investigator-initiated study. EPFC is supported by an Australian National Health and Medical Research Council (NHMRC) Emerging Leadership Investigator Grant (GNT1172873). JJO is supported by an Australian NHMRC Early Career Fellowship Grant (APP1104781). JEB is supported by an Australian Research Council Discovery Early Career Researcher Award (DECRA) Fellowship (DE200100049). All other authors have no conflicts of interest to declare."

We note that you received funding from a commercial source: Gilead Pharmaceuticals

Reviewers' comments:

Reviewer's Responses to Questions

**Comments to the Author**

1. Is the manuscript technically sound, and do the data support the conclusions?

Reviewer #1: Yes

Reviewer #2: Partly

2. Has the statistical analysis been performed appropriately and rigorously? 

Reviewer #1: I Don't Know

Reviewer #2: No

3. Have the authors made all data underlying the findings in their manuscript fully available?

Reviewer #1: No

Reviewer #2: No

4. Is the manuscript presented in an intelligible fashion and written in standard English?

Reviewer #1: Yes

Reviewer #2: No

5. Review Comments to the Author

Reviewer #1: The paper is a significant contribution to our understanding of Asian people’s practices of HIV prevention in Australia. The various cases of HIV-related stigma experienced by Asian gay and bisexual men are convincingly argued and supported by good evidence (i.e. interview accounts, existing scholarship). However, I would like to bring to the authors’ attention the paper’s reductive understanding of Asia. It is surprising that the paper does not take into consideration the ethnic backgrounds of participants despite many of the countries listed are multi-ethnic societies (e.g. Indonesia, Singapore, Malaysia). Because, a Chinese Singaporean might have more in common as a gay or bisexual man with a Chinese Taiwanese or a Chinese Indonesian than an Indian or Malay Singaporean. Often racial struggles are mapped onto and shaped by those of religions. In Malaysia, for example, the sexualities of Malay and non-Malay queer people are governed by two different legal systems. This omission of race and ethnicity not only erases the complexities of the countries studied but also is counterproductive to the authors’ argument that the culture plays a significant role in shaping the participants’ practices of HIV prevention. I hope to see this issue being addressed in the published paper.

Reviewer #2: This is a research paper that explore transnational experiences of Asian-born gay and bisexual men who have sex with men newly arrived in Australia, using minority stress model, intersectionality framework, and social constructionist approach. This issue discussed in this paper is concerning how the experiences of being gbMSM in a relatively gay friendly culture and country may impact on their lived experiences and willingness to test for HIV. I see how the research topic may be valuable, adding to health and sexual health research literature for gbMSM who are immigrants. However, the structure and theoretical framework of this paper may need to be reorganized for clarity, such as the richness of literature review, interpretation/naming of research findings, and implications for sexual health practices. For example, the findings stating that “ingrained fear around HIV testing” is not supported by the data provided. Also, the conclusion is lacking any suggestion of proper culturally or linguistically sensitive services or environment for gbMSM newly arrived in Australia (or Melbourne only?). After reading this manuscript, I also feel that there is information related to blaming Asian cultures and religions underlying, which is extremely dangerous and could be misleading to your readers. This paper would be strengthened if you could reorganize your research findings and draw the conclusion of your findings carefully.

Specific comments are as follows:

Abstract

Page 2

The current version of abstract should be concise and reorganized with the main research findings solely.

The abstract would be improved if you could add on implications for practice and suggestions for future research at the end of the abstract.

Introduction

Page 3

The flow of introduction should be reorganized with more relevant supporting literature on Asian gbMSM who are immigrants in the western countries. For example, the following articles may be helpful for your paper:

Lewis, N. M., & Wilson, K. (2017). HIV risk behaviours among immigrant and ethnic minority gay and bisexual men in North America and Europe: A systematic review. Social Science & Medicine, 179(Complete), 115–128. https://doi.org/10.1016/j.socscimed.2017.02.033

Adams J., Coquilla R., Montayre J., & Neville S. (2019) Knowledge of HIV pre-exposure prophylaxis among immigrant Asian gay men living in New Zealand. Journal of Primary Health Care 11, 351-358.

Neville, S., & Adams, J. (2016). Views about HIV/STI and health promotion among gay and bisexual Chinese and South Asian men living in Auckland, New Zealand. International Journal of Qualitative Studies on Health and Well-Being, 11. https://doi.org/10.3402/qhw.v11.30764

Line 64-73: If you have published your findings in another journal, you still have to add the reference in this paragraph. However, I would suggest removing this whole paragraph from the introduction section in order to concentrate this section on information from the previous studies that can support your research framework. The current flow of the introduction may show that you want to support your framework with your own findings.

Page 4

The literature review focusing on the association between HIV stigma and HIV testing is fine. However, your findings did not clearly support this association.

Materials and Methods

Page 5

The research framework should be reorganized for clarity. The authors mentioned minority stress and intersectionality on Page 3. However, further introducing social constructionist approach on page 5. It seems to me that you have used multiple approaches to explore the studied phenomena. The paper would be strengthened if you could focus on 1-2 approached in this paper. Based on the presentation of your finding, I would suggest removing intersectionality from this paper due to the findings is relevant to social constructionist approach.

Page 6

Line 133

The subtitle indicating “Method, Research Team, and Reflexivity”. I feel confused why you want to introduce the research team here, especially that only two team members are introduced. It would be clearer for your readers if you could focus on method and reflexivity in this section.

The initial of TP on page 6 is different from the same initial on page 42 under contributions (TRP).

Page 8

Data analysis section

I would suggest moving the first paragraph of data analysis section to the data collection section.

Results

I understand that there is no restriction on word count for submitting to PLOS One, however, the journal does encourage authors to present your findings concisely. I would suggest you to present your result section concisely.

Instead of providing so many quotes, it would be helpful for your readers to understand the lived experience of Asian gbMSM in Australia if you could provide a figure addressing their common experiences, process of mentally transition etc.

Page 12

Please provide your rationale why the third theme entitled “Still in a minority group: Experiences of racial discrimination in Australia” cannot be included into the second theme entitled “Living as a gbMSM in Australia”.

Line 239-252. This paragraph sounds to me is an overall conclusion of your research finding. It should be shown at the end of the result section.

Line 254. The theme “Life as a gbMSM in country of origin” is really vague to me. Life in a city or a country could be diverse with positivity and negativity. However, the authors only present negative dimensions of experiences toward their sexuality among Asian gbMSM, it could be very misleading and highly biased.

Page 20

Line 451: The fourth sub-theme “1d. Exposure to HIV-related stigma: ‘HIV is a gay man’s disease’. This theme would be much meaningful and close to your research question if you could spend more space to make the link between HIV stigma and HIV testing in Asia, also parallel to the structure of your second theme (2d). So far, the authors spend more time on describing “HIV is a gay man’s disease”, which is factually and ethically incorrect to statement to make in an academic paper.

Page 24

Line 541. “Life as a gbMSM in Australia”- This theme talks about lived experiences of Asian gbMSM in Australia, same as the firs theme, this is really a vague theme for me.

Those quotes related to acceptance and freedom may present part of lived experiences in Australia, other experiences related to racism and xenophobia may cause different experiences. I would suggest the authors to reorganize your themes.

The findings would be interpreted meaningfully and see the whole picture of Asian gbMSM in Australia if the authors could reorganize your themes beyond personal level perspective. In your quotes, for instance, the authors have found that participants would choose to come out to their friends if the environment is gay friendly and accepting. This information to me is related to interplay between personal, interpersonal, and community level factors. Other quotes also describe social support was associated with coming out among Asian gbMSM.

Page 30.

“2d. Ingrained fear around HIV testing” Please review your quotes under this section carefully and check whether if there is a connection between sociocultural environment and fear of getting an HIV test. The current quotes for me do not support your statement on “many described residual or ingrained fear of HIV testing in Australia as a result of the sociocultural environment in which they grew up, where HIV was heavily stigmatised.” Instead, All the quotes presented in this section are describing their worries and feelings of getting an HIV test. Are those worries a result of that they experienced in their home countries? It would be clear to your reader if you could carefully explain what you concluded.

Also, quotes in this section actually addressed a very good point but is not developed further by the authors: The reason why they delayed having an HIV test was because they may not know where they can be tested and they don’t know if the sexual health clinics are friendly to racial minorities at the beginning. This statement/point is one of the potential findings that you could explore more.

Page 33

Theme 3 is a very interesting topic and could contribute to the current literature on the topic of racial discrimination among gbMSM. I would suggest authors to independently produce a manuscript for their experiences.

Discussion

Page 39

The discussion section would be much meaningful if you could provide implications for sexual health practice based on your research findings.

Line 920-923: “Many men in this study expressed continued anxiety about getting tested for HIV after arriving in Australia due to the negative experiences and HIV-related stigma pervasive in their countries of origin and for some men this resulted in delayed HIV testing” It would be clear to your readers if you make such connect/statement based on your research finding. To me, I did not see the reason of delayed HIV testing was associated with stigma pervasive in their countries of origin. To my understanding, Asian gbMSM in Melbourne may be lacking of HIV testing information to access services.

Line 938-950. If the authors would be interested in writing another paper on the topic of lived experienced of racial discrimination among Asian gbMSM, this paragraph should be removed.

In limitation, authors address that this paper “was not to provide a generalizable findins applicable to all gay or bisexual men born in Asia and living in Australia”. However, you did try to generalize your participants’ experiences in Melbourne to Australia throughout the paper. This issue would be improved if you could switch those parts using “in Australia” to “in Melbourne” to decrease your underlying generalization.

6. PLOS authors have the option to publish the peer review history of their article (what does this mean?). If published, this will include your full peer review and any attached files.

Reviewer #1: No

Reviewer #2: No

---

## [Author Response · Author response to Decision Letter 0]

20 Aug 2020

Editorial Requests and Comments

Clearly the reviewers' individual recommendations vary; however, what is common to both reviewers, not incidentally both of whom occupy subject positions perhaps more aligned with the participants than the two apparent main authors who conducted the interviews and data analysis--not necessarily a critique, but a relevant observation--is a strong questioning of several presumptions that underlie the text and the analysis. These coalesce in the apparent homogenization of what are vastly different Asian cultures, ethnicities, religions and participants, and what emerges as a characterization of these men's 'home' cultures as unilaterally stigmatizing, homophobic, and seemingly antithetical to gay men's mental health. On the other hand, the description of methods and some of the analysis appears very thoughtful, supported by rich data, and guided by theory--all of which are strengths of the manuscript. And both reviewers indicate the import of the subject and strong potential for meaningful contribution to the literature.

Response: Thank you for your helpful comments and for your consideration of this manuscript. We believe we have addressed and amended, where possible, each of the concerns and comments from yourself and the reviewers below. Where text has been changed we have included it below in blue. 

Editor, Comment 1

1. When you discuss reflexivity, please consider the subject positions of the apparent lone interviewer and data analysis persons; how might these impact on what the participants reveal in the interviews; what questions are asked; how the data are analyzed; conceptualizations of their 'home' countries; and the potential for a somewhat monolithic treatment of the construct of "Asia" and "Asian gay men"? This is in no way to say that only same-ethnicity, same-gender, etc. researchers can do valid research with like participants; but as the authors seem to be aware, and particularly in constructivist qualitative approaches, their identities do matter; and these need to be addressed in more depth. It is also generally a weakness in qualitative inductive approaches to have data analysis almost entirely conducted by 1 person--please note this as an additional limitation;

Response: Thank you for this valuable feedback. We have added further detail about our process of reflexivity in both the methods and the limitations sections of the manuscript, specifically addressing the diversity of our team and their relevant backgrounds to further inform the readers where our preconceived ideas with regards to the subject matter predominately originate from, as well as noting the limitations of having two white female researchers leading the analysis. The following clarification has been made to the methods section:

(Page 6, lines 148-163)

“Semi-structured interviews were chosen for this study as they allowed men to share their lived experience and personal reality of being a gbMSM. The interview schedule was jointly designed by a majority of the research team, including NM (FAChSHM, PhD), a male clinical epidemiologist and consultant HIV physician; EPFC (PhD) a male epidemiologist with several years’ experience in sexual health; JJO (FAChSHM, PhD), a male sexual health physician and researcher with a special interest in increasing access to sexual health services to marginalised populations; JEB (PhD), a female senior research fellow with a doctorate in public health who specialises in social research in the area of sexual and reproductive health; and TRP (PhD), a female research fellow with several years’ experience working in sexual health. NM and JO particularly had extensive quantitative research experience describing trends in HIV diagnoses within newly-arrived Asian-born gbMSM. The team members have combined decades of experience in this research area, have diverse sexual identities and cultural backgrounds, including two members of Asian ethnicity. Thus the nature of the questions chosen were influenced by the clinical and cultural experiences of the researchers themselves, which in turn will have impacted on the meanings derived from the interviews.

Interviews were conducted by TRP. Participants had no prior relationship with TRP and were informed that the study was being undertaken to understand their experiences around HIV testing and preferences for HIV prevention strategies in light of rising rates of HIV diagnoses in this population.” 

We have added this sentence to the beginning of the Data Analysis section (lines 228-232):

“The research team met regularly to discuss the findings from the study, which involved reflecting on the research methodology and identifying areas of improvement within the interview structure as well as reflexively examining and challenging our underlying perspectives about the research participant’s attitudes and cultural influences (1).“

Furthermore, we have now addressed reflexivity with regards to analysis of the data in more depth in the limitations, as outlined below (Lines 1238-1260): 

“Another limitation that should be considered is that the interviews were all conducted by one researcher (TRP), a white female. Participants may have been reluctant sharing information with someone they did not feel was a peer or who could not fully understand their lived experience as an Asian born gbMSM. Further to this end, TRP and JEB (another white female) were responsible for the majority of the data analysis, with TRP doing the crux of the analysis and JEB conducting cross-checking on a sub-set of transcripts to confirm the coding framework and thematic analysis. One of the risks with qualitative research is that researchers can unintentionally bias various aspects of a study, with their own beliefs, values or preconceptions, which can result in an ‘outsiders’ view or interpretation of the data (2). Despite TRP and JEB meeting regularly with the wider research team to discuss findings and challenge their assumptions about the attitudes and cultural backgrounds of the participants, it is possible the data were interpreted through their own cultural and gender lens as white female researchers. Additionally, had two researchers coded the entirety of the data (cross-coding/multiple coding) instead of one researcher doing the majority of the analysis, this would have provided more rigor to the analysis. While ideal, this is generally prohibitive in qualitative research due to cost and effort, when compared with the standard convention of cross-checking by a researcher in a supervisory role (3).”

References: 

1. Barrett A, Kajamaa A, Johnston J. How to be reflexive when conducting qualitative research. Clin Teach. 2020;17(1):9-12.

2. Birt L, Scott S, Cavers D, Campbell C, Walter F. Member Checking: A Tool to Enhance Trustworthiness or Merely a Nod to Validation? Qual Health Res. 2016;26(13):1802-11.

3. Barbour RS. Checklists for improving rigour in qualitative research: a case of the tail wagging the dog? BMJ. 2001;322(7294):1115-7.

Editor, Comment 2

2. Are there no strengths revealed that emerge from an of the 'home' cultures? Any redeeming value of the experience of growing up as a gay man in these countries? I would encourage the authors to critically assess how their portrayal of Asia, while unintentional, may be read by others, in a rather flat, uniform, and quite negative context for all gay men; 

Response: Thank you for this comment, which we have given the utmost of consideration. It is very important to us that we are not portraying Asian born gbMSMs experiences as flat, uniform or negative. It is worth reiterating here that the main aim of this research study was to explore strategies that newly-arrived to Australia Asian born gbMSM were using or preferred to use to prevent HIV infection, in light of rising rates of HIV in this population compared to rates of new HIV in other GBMSM living in Australia. This study was therefore designed to explore participant’s knowledge levels about HIV prevention strategies, and what may have shaped their knowledge levels. We were particularly interested in connections they had made in Australia as we know from the literature that peer connections are a facilitator of sexual health. While we did not expect all of our participants to have extensive knowledge of HIV strategies, we did consider in designing this study that by recruiting from a sexual health clinic our participants would already have some access to sexual health services. To that end, we asked participants some variation of “What has being a gay man in Australia been like for you?” near the beginning of each interview—for example, here is how we asked this question in the very first interview for this study: 

Researcher (TRP): So, what is your experience living as an Asian-born, gay man in Australia?

Participant: Uhhh it was nice, and uh, experience. There is a lot of freedom here, and you can be gay… back in Pakistan, you cannot be openly gay.

The second participant in the study responded to this question similarly:

Researcher (TRP): what is your experience of living as an Asian-born, gay man here in Australia? 

Participant: Uh, I would say that in Australia, we’re accepted, and then there’s no, you, you can’t feel any difference between you and the normal straight person over the way it is. Well back in Malaysia, there’s a boundary, when you, you know where’s the line between you and, like you can’t do so much things back in Malaysia, because if they don’t say that it’s legal and they don’t say it’s illegal. Yeah.

Most of the participants ended up answering this question this way, in other words, used this question as a starting point to describe differences in acceptability of their sexual identity in their countries of birth versus in Australia. Thus, while we did not design the survey specifically to compare and contrast their experiences of acceptance of their sexual identity between their countries’ of birth and Australia, participants almost always brought this up in response to this question and we explored it further from that point. Throughout the analysis, and in conjunction with routine discussions of the data with the research team, it became apparent that to publish a manuscript solely outlining the strategies our participants used and preferred for staying HIV negative (including facilitators towards sexual health such as peer connections, which are addressed in our second manuscript) that did not lend a voice to these experiences of the internalised stigma that these men were reporting, would be remiss and misleading. It was never our intention to suggest that these men had only negative experiences in their countries of origin, nor did we view or want to portray them as victims. We acknowledge that our exploration of their experiences in their countries of origin only with regards to these topics has resulted in their countries of origin being portrayed in a negative light. We have tried to address this in the limitations with the following addition (lines 1204-1217): 

“An important limitation to this study is that the experiences described by the participants in this study were centered around their sexual identity and attitudes towards HIV, and thus their responses are not indicative of their entire lived experience in their countries of origin, nor their holistic attitudes towards living in their countries of origin. The primary aim of the study was to explore strategies that newly-arrived to Australia Asian born gbMSM were using or preferred to use to prevent HIV infection, in light of rising rates of HIV in this population. This study was therefore designed to explore participant’s knowledge levels about HIV prevention strategies, and what may have shaped their knowledge levels. In asking men about their knowledge levels we asked men what it was like to live as a gbMSM in Australia compared to their home country whereby they commonly described differences in acceptability of their sexual identity in their countries of birth versus in Australia. Given the topics that were explored and arose, the results of our study may lend themselves to more negative aspects or experiences of their countries of origins.” 

Furthermore, in an effort to reduce the depiction of the countries represented as uniform, we have restructured and added to the ending of section 1a. Laws, religion, tradition and culture to include participant’s beliefs that their regions have different attitudes towards homosexuality than the country as a whole or other cities within the country (lines 372-424):

“A few participants discussed differences in views of homosexuality in their local regions versus other cities in their countries of origins or the country of origin as a whole. One participant described the interplay of different communities’ (religious, social, lifestyle) attitudes towards homosexuality in his small town in East Malaysia. To him, these differing community attitudes resulted in homosexuality being viewed as a sort of open secret—people living in the area know gay people are there and while they don’t condone expressions of homosexuality, they also don’t chastise them for their lifestyle. 

But in terms of, like, community wise, so for where I’m from, specifically the state of [state named] so that’s in East Malaysia so there’s sort of this open secret, where it’s, like, ‘oh, that clique of people, they’re openly gay’, but no one will talk about them. There’s no need to chastise them for their lifestyle, but we all know they are there, that kind of thing. I guess it’s just because of – just certain – I don't know what it is, but I guess it’s just like, sexual exploration is not really much of a thing back there, so yeah… And Malaysia is a Muslim majority country is that you just don’t talk about homosexuality, as a general rule, but yeah, that’s sort of the weird dynamic I was saying – like, yeah, but most – at least from my hometown, is Muslim. They’ve taken weird stances on homosexuality… like things like you can be gay but you don’t have to express it. I’m not sure how to respond to that, so that’s part of the restrictions within the community I was talking about. It’s not even just the gay community, it can be the Muslim community, it can be within – because my hometown is a relatively small town, so I’d say people from the same high school, or the same colleges or uni[versity], they tend to know each other, so that’s sort of a community within itself, yeah.

--Participant 11, Malaysia, 4 years in Australia

Another participant spoke of his society’s conservative values even restricting what clothes people can wear. To him, being openly gay in his country of origin was not fathomable since they cannot even accept men wearing shorts. He did not place the responsibility for these rules with his religion, but rather with the society and culture in his ‘Old City’, which he depicted as being more traditional than other cities in India. 

I’m from the Old City. It’s just that I don’t blame my religion but then I blame the society and the culture. They are the ones that are setting the rules. The religion has never set the rules…I remember this one day I went out and my friend picked me up. I was lucky. I was like, “All right. I can just go quickly.” I went down and sat in the car but then on my way back I was dropped somewhere a little bit far away, like a five-minute walk to my house. I cannot forget that five-minute walk, the names I was called and the looks I got were horrible … Forget being gay, I can’t wear shorts. No straight man can wear shorts in my country. Not my country, just my particular society. It’s so weird. Being gay would probably be the end of me.

 –Participant 6, India, 5 months in Australia”

In line with this change, we have amended the discussion of this limitation as follows (Lines 1222-1228):

“Further to this limitation, “Asian-born” is an umbrella term applicable to a vast array of diverse ethnicities, societies and cultures, a fraction of which were discussed here. With regards to the topics presented here, there are, in addition to the variations between countries, variations in religions, cultures and attitudes within countries, as our participants touched on. Our aim was not to provide a generalizable finding applicable to all gay or bisexual men born in Asia and living in Australia, but rather to capture the depth and breadth of experiences from people in what has been identified as a HIV vulnerable population in order to further direct targeted HIV prevention in this group. Given the sparsity of research into Asian-born gbMSM migrants living in Australia, the diversity of our study may in fact be considered a strength as our sample included men from a broad range of Asian countries, of varying ages and education levels. By focussing on newly arrived gbMSM we have been able to include narratives from several countries that have differing family, religious and cultural influences.”

Editor, Comment 3

3. What of the strengths and resilience of the participants? Arguably the weight of the narrative construction presented here is that of 'victims' of deeply internalized stigma from their home countries and cultures, and additional stigma received in Australia. For ex., how do some or most Asian gay men immigrants to Australia not contract HIV? Qualitative research also can use 'negative case examples' as a means to shed more insight and more nuances in the data so as to avoid universalistic constructions, particulalry of 'victims'. This would help to provide a more full picture of participants as also possesing agency and strength; 

Response: The topic of strength and resilience is important, and we certainly want to present men’s experiences without portraying them as victims with no agency. We discuss facilitators to sexual health in our separate paper from this study, however we have made a few amendments to this manuscript to better reflect the positive connections men have in their countries of birth. We have added to the 1c. Repressed and hidden identities section men’s experiences in their countries of origin of having peer support. While we previously stated that some men were open to friends but not family members, as one of the reviewers also pointed out, we did not elaborate on those men’s experiences. We agree that given the importance of social connections it is best to expand upon this topic as we have now done (Lines 565-610): 

“Some participants described having a supportive group of friends who were also gay in their countries of origin to whom they felt comfortable being open about their sexual identity. For these men, having gay friends gave them a sense of strength in numbers and the courage to be themselves in the face of judgement. 

So back in high school my friends were - I surrounded myself with gay friends actually … even the teacher makes fun [of us]. They counsel us. Because it's a boys' school you have to be manly because it's stated in the [motto] of the school… it's a great school, I didn't complain… I remember going to the counsellor room and tell us ‘why you want to do this, what would your parents think?’…[about us] being feminine …we try to bring a message [that] even though we are feminine, but we have discipline. We have the courage that you want to be. We show them all. So I'm kind of proud of that one. 

–Participant 20, Malaysia, 3 years in Australia

Yes I would say I was a part of the gay community in Singapore because partly because Singapore is so small so like if you’re gay you know who’s who, that kind of thing. It’s kind of nice because the population, the community is quite well knit. We have a lot of events and we always see the same few faces and just say hi and so on yes… Yes, that’s a nice thing to see in Singapore…I guess like I got to know more gay people through those events, because even though like, how do you say this, I’ve not made any gay friends through school or work or anything…I don’t usually say out that I’m gay kind of thing back in Singapore….

–Participant 16, Singapore, 3 months in Australia

Other participants spoke about being open about their sexual identity to close friends in their countries of origins and not just those that were also gay. One participant explained how important it was for him to be open with his friends in China in order to feel like himself. For this man, having a boyfriend gave him the courage to be open to his friends about his sexual identity despite feeling that people may see him as ‘abnormal’. 

I got the courage because I have a boyfriend that time, so I got the courage to come out to my friends…The real reason is I want to come out because staying in the closet, it's a very hard life, I should say. You should pretend like you are like not a gay and they will think, ‘oh, you're just like the other people who like girls.’ But I don't want that. I want my friends [to] know truly about me… It's not normal [being gay], yes, to most of the people. They don't think it's normal things, but I think it's normal, we are… I think it's—after I coming out—it's a little, much easier. I don't pretend to be what these other normal people… I don't want to pretend to be someone. I just want them to [see] myself. Truly who I am is to be honest to my friends and to myself.

—Participant 17, China, 3 years in Australia”

Further to this end, we have also added a paragraph in the Discussion that highlights some participant’s social connections and discusses the literature on peer support and resilience (Lines 1109-1126): 

“While men often spoke of hiding their sexual identity to family members, it is important to note that there were some that were open to friends in their countries of origins, including a couple of men who discussed the strength they found from having friends in their countries of origin who were also gay. While we did not measure resilience (i.e. the ability to recover after experiencing adversity) among our participants, peer support is a known protective process that leads to the development of resilience (4, 5), which in turn may have a positive effect on mental health and well-being (6). In our separate paper from this study, we discuss our findings that social support was a facilitator to sexual health within this group, notably some participant’s positive experiences with local LGBTQI organisations that offer peer-led social support workshops. Increasing awareness of and opportunities to engage with such services may be beneficial for men in this group in terms of bolstering their social support in Australia. Additionally, it is important for these services to have an awareness of the stigma and discrimination Asian born gbMSM can experience in Australia to enable them to factor this into culturally appropriate services which meet their needs. Further research could be done to examine mental health, resilience, and social support within this group and the implications for HIV prevention.”

Additionally, we have now highlighted in the Discussion the self-compassion many of our participants showed in their descriptions of discrimination experienced in Australia, which other studies have identified as a component of resilience among people at the intersect of marginalised populations (Lines: 1162-1175):

“However, it is important to highlight some of the participant’s responses to racial discrimination in Australia, notably how many explained that they do not take sexual racism on dating apps personally, but rather see it as men having preferences. Similarly, several men when describing instances of discrimination, saw the experience as an indictment of the perpetrator of the discrimination and not themselves, for example rationalising that the perpetrator was having a bad day or lacked cultural understanding. These responses to racial discrimination indicate a level of self-compassion among the participants (7). Previous research among students in the USA suggested that self-compassion may play an important role in coping with stigma among those students who were in both a sexual and racial minority (8). Future research could investigate the level of self-compassion within this population and its effects on resilience in the face of discrimination.”

References: 

4. Luthar SS, Sawyer JA, Brown PJ. Conceptual issues in studies of resilience: past, present, and future research. Ann N Y Acad Sci. 2006;1094:105-15.

5. Handlovsky I, Bungay V, Oliffe J, Johnson J. Developing Resilience: Gay Men's Response to Systemic Discrimination. Am J Mens Health. 2018;12(5):1473-85.

6. McLaren S, Jude B, McLachlan AJ. Sense of Belonging to the General and Gay Communities as Predictors of Depression among Australian Gay Men. International Journal of Men's Health. 2008;7(1):90-9.

7. Raes F, Pommier E, Neff KD, Van Gucht D. Construction and factorial validation of a short form of the Self-Compassion Scale. Clin Psychol Psychother. 2011;18(3):250-5.

8. Vigna AJ, Poehlmann-Tynan J, Koenig BW. Does self-compassion covary with minority stress? Examining group differences at the intersection of marginalized identities. Self and Identity. 2018;17(6):687-709.

Editor, Comment 4

4. What might it say that all participants "declined" member checking? While one cannot lay 'blame' on the researchers, it is at least worthwhile to conjecture about what might have created a more welcoming or inclusive approach to the research that at least a few participants would indicate willingness to re-engage? It also seems as if the authors adopt a very narrow view of member checking. Were participants invited to offer their own interpretations of the data? or only to review the apparent accuracy of the transcripts? This is quite important as a successful approach to member checking might have helped to 'correct' what both reviewers read as a rather 'outsider' conflation and construction of "Asian culture" and "Asian gay men". This is also a limitation of the research that needs to be identified

Response: Thank you for this important insight. We have considered what it could have meant for the study to have offered a more collaborative form of member checking to the interpretation of the data and the rigour of the analysis. We have amended the limitation section of the Discussion to include a more thorough critique of our not having done this, including the potential for greater rigour in our analysis of these men’s experiences and minimising potential bias from our outsider’s lens (Lines 1260-1273). 

“A further limitation of this study was that participants were only offered a relatively narrow form of member checking, whereby they were asked if they would like to check the accuracy of their manuscript. While all participants indicated they were happy to be contacted again following the interview, none took up the offer to check their manuscripts for accuracy. In hindsight, having a broader approach to member checking, whereby men were re-engaged at the point of data interpretation to participate in a member check interview to offer their views on interpretation or their own interpretations of the data, would likely have ensured a more rigorous and potentially more accurate understanding of the data, thereby minimising the risk of a ‘outsider’ conflation or construction of Asian culture and Asian gbMSMs experiences. If broader member checking activities had been employed, promoting a more inclusive approach to the research, men would likely have re-engaged at a greater level, allowing for a more nuanced understanding of their experience”.

Editor, Comment 5

5. Importantly, you must provide a full citation for the other article noted as one in which some of the data have already been published; this is imperative both to assure reviewers and readers (and editors) that the articles are not redundant, and also for you to briefly but clearly indicate how the present manuscript is different from the other manuscript mentioned based on the same study

Response: The subsequent manuscript from this study was recently submitted to PLOS ONE and a response to the editor is currently being drafted that outlines the differences in the two papers and justifies the separate manuscripts. While these manuscripts compliment each other, we feel they are stand-alone manuscripts in their own right. We have made an additional statement about the findings from our separate paper in the Discussion (Lines 1116-1119):

“In our separate paper from this study, we discuss our findings that social support was a facilitator to sexual health within this group.”

Additionally, the overall study aim is stated in the Introduction (lines: 74-80) and we now have reiterated the study aim in the Discussion to add context for these results (lines 1208-1215):

“The primary aim of the study was to explore strategies that newly-arrived to Australia Asian born gbMSM were using or preferred to use to prevent HIV infection, in light of rising rates of HIV in this population. This study was therefore designed to explore participant’s knowledge levels about HIV prevention strategies, and what may have shaped their knowledge levels. In asking men about their knowledge levels we asked men what it was like to live as a gbMSM in Australia compared to their home country whereby they commonly described differences in acceptability of their sexual identity in their countries of birth versus in Australia”.

Editor, Comment 6

6. As reviewer 2 notes, and I concur, it would be helpful to identify potential implications for HIV testing education, outreach, services, providers in Melbourne. It might be read as if the the full responsibility is on these individuals to navigate what may HIV services as fixed entitites. How might such services recognize the stigma and discrmination within the Australian context and create more culturally appropriate services to meet participants needs and encourage earlier HIV testing?

Response: We discuss this in-depth in our second manuscript but have now also included this statement in the Discussion (Lines 1116-1126):

“In our separate paper from this study, we discuss our findings that social support was a facilitator to sexual health within this group, notably some participant’s positive experiences with local LGBTQI organisations that offer peer-led social support workshops. Increasing awareness of and opportunities to engage with such services may be beneficial for men in this group in terms of bolstering their social support in Australia. Additionally, it is important for these services to have an awareness of the stigma and discrimination Asian born gbMSM can experience in Australia to enable them to factor this into culturally appropriate services which meet their needs. Further research could be done to examine mental health, resilience, and social support within this group and the implications for HIV prevention.

Editor, Comment 7

7. I would suggest that while it is plausible that the sample in recruiting from sexual health clinics identified the most 'willing' of participants to engage with services, such that other Asian gay men may be every more 'vulnerable', is it not also plausible that some Asian gay men who are less 'integrated' into Australian institutions possess cultural capital and cultural strengths that are protective factors against stigma and discrimination? Indeed this is found in a good deal of research on immigrants in many majority cultures.

Response: Thank you for this comment. We agree that the buffering effect of ethnic identity on discrimination-related stress is a very interesting topic that could be further explored in this group. We also want to reiterate that a very real barrier for migrants with regards to HIV testing is not speaking the host- country language. We have further clarified this in the limitations so that our language is clear that we are referring to “HIV vulnerability” and not overall “vulnerability” and have added a reference to support this statement. We then discuss the interesting research with regards to ethnic identity and discrimination:

“Thus, men who are not connected to sexual health services or those with limited English have not had their voices heard in this study. It is possible that members of these groups may be even more vulnerable to HIV due to their further marginalised status, as not speaking the host countries’ language has been a reported barrier to HIV testing for migrants in high-income countries (9). Additionally, these men were willing to discuss topics with the researcher that they felt were taboo in their countries of birth and therefore they may be more open than other newly-arrived Asian-born gbMSM. This study did not measure the levels of ethnic identity among the participants, but recent research has questioned the role of ethnic identity as a protective buffer for the stress of discrimination (10). Future research could investigate whether men in this group with stronger ethnic identity have ameliorated stress levels in the face of discrimination.”

References:

9. Blondell SJ, Kitter B, Griffin MP, Durham J. Barriers and Facilitators to HIV Testing in Migrants in High-Income Countries: A Systematic Review. AIDS Behav. 2015;19(11):2012-24.

10. Mossakowski KN, Wongkaren T, Hill TD, Johnson R. Does ethnic identity buffer or intensify the stress of discrimination among the foreign born and U.S. born? Evidence from the Miami-Dade Health Survey. J Community Psychol. 2019;47(3):445-61.

Editor, Comment 8

8. The authors admirably note use of the COREQ guidelines. It would strengthen the submission to attach the COREQ checklist and indicate where in the manuscript each of the guidelines were addressed. Not every manuscript satisfies all of the items delineated, but evidence to show how the authors met many of these guidlines would be very helpful. This also relates to comment 1, which requires a fuller and perhaps more earnest engagement with reflexivity. For ex., see: Tufford, L., & Newman, P. (2012). Bracketing in Qualitative Research. Qualitative Social Work, 11(1), 80–96. https://doi.org/10.1177/1473325010368316

Response: Thank you for this suggestion, we have now completed the COREQ checklist to show page numbers for the items in the checklist and have attached the checklist to this submission. 

Editor, Comment 9

9. Finally, while the authors need to address the core concerns shared across the reviewers, and as described above, I do not necessarily expect that each and every comment of the 2nd reviewer needs to be acceded to. However, if there is some misinterpretation by the reviewer in certain cases, I would ask the authors to briefly acknowledge and explain it. In other cases, the revisions as suggested should be incorporated. 

Response: Thank you for this clarification, we have addressed all concerns from both reviewers item-by-item below. 

Formatting Amendments

Formatting, Comment 1

Response: We have now amended the title page and changed the reference style from Vancouver to PLOS as directed. 

Formatting, Comment 2

2. Thank you for including your competing interests statement; "NM, EPFC and JJO have received a research grant from Gilead Pharmaceuticals to conduct this investigator-initiated study. EPFC is supported by an Australian National Health and Medical Research Council (NHMRC) Emerging Leadership Investigator Grant (GNT1172873). JJO is supported by an Australian NHMRC Early Career Fellowship Grant (APP1104781). JEB is supported by an Australian Research Council Discovery Early Career Researcher Award (DECRA) Fellowship (DE200100049). All other authors have no conflicts of interest to declare."

We note that you received funding from a commercial source: Gilead Pharmaceuticals

Response: We have amended the competing interest statement as requested:

"NM, EPFC and JJO have received a research grant from Gilead Pharmaceuticals to conduct this investigator-initiated study. Gilead Pharmaceuticals had no input in the research plan, recruitment, analysis or decision to publish the paper. EPFC is supported by an Australian National Health and Medical Research Council (NHMRC) Emerging Leadership Investigator Grant (GNT1172873). JJO is supported by an Australian NHMRC Early Career Fellowship Grant (APP1104781). JEB is supported by an Australian Research Council Discovery Early Career Researcher Award (DECRA) Fellowship (DE200100049). All other authors have no conflicts of interest to declare. This does not alter our adherence to PLOS ONE policies on sharing data and materials."

Formatting, Comment 3

Response: Due to the small sample size and the interview transcripts containing sensitive and potentially identifying information, the ethics committee have not approved public release of this type of data. Interested researchers may contact Emily Bingle at the Alfred Hospital Ethics Committee if they would like access to the data: research@alfred.org.au, quoting project 222/19. 

Reviewer 1

The paper is a significant contribution to our understanding of Asian people’s practices of HIV prevention in Australia. The various cases of HIV-related stigma experienced by Asian gay and bisexual men are convincingly argued and supported by good evidence (i.e. interview accounts, existing scholarship). However, I would like to bring to the authors’ attention the paper’s reductive understanding of Asia. It is surprising that the paper does not take into consideration the ethnic backgrounds of participants despite many of the countries listed are multi-ethnic societies (e.g. Indonesia, Singapore, Malaysia). Because, a Chinese Singaporean might have more in common as a gay or bisexual man with a Chinese Taiwanese or a Chinese Indonesian than an Indian or Malay Singaporean. Often racial struggles are mapped onto and shaped by those of religions. In Malaysia, for example, the sexualities of Malay and non-Malay queer people are governed by two different legal systems. This omission of race and ethnicity not only erases the complexities of the countries studied but also is counterproductive to the authors’ argument that the culture plays a significant role in shaping the participants’ practices of HIV prevention. I hope to see this issue being addressed in the published paper.

Response: Thank you for this comment, which was also mentioned similarly by the editor. We did ask each participant if they identified with any particular ethnicity or religion and we have made this more clear in the demographics table now by adding ethnicity when a participant gave one. Men often brought up religious influence on the attitudes toward sexual identity and HIV, which we have mentioned in the manuscript, and sometimes mentioned regional differences which we have now added in light of yours and the editor's comments, specifically in section 1a. Laws, religion, tradition and culture. None of our participants brought up differences in how HIV is viewed by ethnicities in their countries of origins, but we also did not ask specifically about this. In addition to the changes we have made above to address the editors concerns around this issue, we have now also amended this limitation in the Discussion (Lines 1218-1222):

“Similarly, while we asked participants if they identified with a particular ethnicity or religion, we did not specifically ask about perceived differences between ethnicities or religions in their countries of birth with regards to attitudes toward sexual identity and HIV. Further research is warranted to better understand how ethnic identity and religion influences attitudes toward sexual identity and HIV among subcultures in different Asian countries”

Reviewer #2

This is a research paper that explores transnational experiences of Asian-born gay and bisexual men who have sex with men newly arrived in Australia, using minority stress model, intersectionality framework, and social constructionist approach. This issue discussed in this paper is concerning how the experiences of being gbMSM in a relatively gay friendly culture and country may impact on their lived experiences and willingness to test for HIV. I see how the research topic may be valuable, adding to health and sexual health research literature for gbMSM who are immigrants. However, the structure and theoretical framework of this paper may need to be reorganized for clarity, such as the richness of literature review, interpretation/naming of research findings, and implications for sexual health practices. For example, the findings stating that “ingrained fear around HIV testing” is not supported by the data provided. Also, the conclusion is lacking any suggestion of proper culturally or linguistically sensitive services or environment for gbMSM newly arrived in Australia (or Melbourne only?). After reading this manuscript, I also feel that there is information related to blaming Asian cultures and religions underlying, which is extremely dangerous and could be misleading to your readers. This paper would be strengthened if you could reorganize your research findings and draw the conclusion of your findings carefully.

Response: Thank you for your important insight and comments about our manuscript. We have addressed the concerns that you raise step by step below. 

Reviewer 2, Comment 1

1. Abstract, Page 2: The current version of abstract should be concise and reorganized with the main research findings solely.

Response: Upon review, we feel that the abstract adequately summaries the main research findings and does not warrant restructuring. We have adhered to PLOS One’s guidelines for abstract structure and length and have included only content we feel is relevant to the main study findings. We have added a line about hiding sexual identities to reflect the amended text that some men spoke of being open to friends in their countries of origin (Lines 48-51): 

“Participants described hiding their sexual identities in their country of origin, particularly from family members, due to fear of judgement and discrimination resulting from exposure to sexual identity and HIV related stigma in their countries of origin, although some were open to friends.”

Reviewer 2, Comment 2

2. The abstract would be improved if you could add on implications for practice and suggestions for future research at the end of the abstract.

Response: Thank you for this comment, we have now amended the conclusion of the abstract with the following additional sentences on implications from our findings (Lines 59-64):

“Our data highlights the potential discrimination Asian-born gbMSM face in Australia, which has implications for social connectedness, particularly with regard to LGBTQI communities and HIV testing practices. Future studies should determine effective strategies to reduce sexual identity and HIV-related stigma in newly-arrived Asian-born gbMSM”.

Reviewer 2, Comment 3

3. Introduction, Page 3: The flow of introduction should be reorganized with more relevant supporting literature on Asian gbMSM who are immigrants in the western countries. For example, the following articles may be helpful for your paper:

• Lewis, N. M., & Wilson, K. (2017). HIV risk behaviours among immigrant and ethnic minority gay and bisexual men in North America and Europe: A systematic review. Social Science & Medicine, 179(Complete), 115–128. https://doi.org/10.1016/j.socscimed.2017.02.033

• Adams J., Coquilla R., Montayre J., & Neville S. (2019) Knowledge of HIV pre-exposure prophylaxis among immigrant Asian gay men living in New Zealand. Journal of Primary Health Care 11, 351-358.

• Neville, S., & Adams, J. (2016). Views about HIV/STI and health promotion among gay and bisexual Chinese and South Asian men living in Auckland, New Zealand. International Journal of Qualitative Studies on Health and Well-Being, 11. https://doi.org/10.3402/qhw.v11.30764

Response: Thank you for highlighting these sources for us. We have referenced the last two papers in our separate paper from this study as they more closely support HIV and PrEP knowledge which is the topic of our separate paper. We have now included reference to the first paper you’ve mentioned in the introduction (Lines 91-95): 

“A previous systematic review of migrant minority gbMSM in the USA and Europe found varying HIV risk profiles in migrant gbMSM depending on their ethnicity, country of origin, and current location and highlighted the importance of viewing HIV risk in the context of migration (11)”

Reference:

11. Lewis NM, Wilson K. HIV risk behaviours among immigrant and ethnic minority gay and bisexual men in North America and Europe: A systematic review. Soc Sci Med. 2017;179:115-28.

Reviewer 2, Comment 4

4. Line 64-73: If you have published your findings in another journal, you still have to add the reference in this paragraph. However, I would suggest removing this whole paragraph from the introduction section in order to concentrate this section on information from the previous studies that can support your research framework. The current flow of the introduction may show that you want to support your framework with your own findings.

Response: Thank you for this suggestion, however, given the suggestion from the editor to be clearer in what the previous paper outlines, we feel it is best to keep this paragraph. Additionally, it is important for the reader to understand that this manuscript presents findings that were tangential to the primary aim of the study. As we have clarified in this version (and see our response to the Editor, Comment 2) here we present ‘additional’ data that emerged from the study, which is important to note given our participants are only speaking about one aspect of their lives in their countries of origin and not a holistic view of their lives in their countries of origin. 

Reviewer 2, Comment 5

5. Page 4: The literature review focusing on the association between HIV stigma and HIV testing is fine. However, your findings did not clearly support this association.

Response: Thank you for pointing this out, we have added more to the quotes to better describe the context of delayed testing due to HIV stigma, please see your Comment 19 below where you raise this issue again for more detail and the amended Results from line 860-898. 

Reviewer 2, Comment 6

6. Materials and Methods, Page 5: The research framework should be reorganized for clarity. The authors mentioned minority stress and intersectionality on Page 3. However, further introducing social constructionist approach on page 5. It seems to me that you have used multiple approaches to explore the studied phenomena. The paper would be strengthened if you could focus on 1-2 approached in this paper. Based on the presentation of your finding, I would suggest removing intersectionality from this paper due to the findings is relevant to social constructionist approach.

Response: A social constructionist approach was the qualitative framework that guided the study. From this viewpoint, people’s perceptions of reality and the meaning they give to their experiences are shaped by the social and cultural norms operating at the time and within that context. It is a qualitative framework that it entirely appropriate to the topic of investigation and the methods utilised. 

The minority stress model assists in explaining the effects of minority sexual identity on mental health and posits that marginalised groups experience more social stress than non-marginalised groups: an intersectionality occurs when their ethnicity and migration status is also considered. These models or approaches can help us understand what may be contributing to HIV vulnerability in Asian born-MSM, and lend themselves or align well with a social constructionist framework. As such, we feel each makes their own contribution independently to the paper. We are reluctant to make changes in this regard.

Reviewer 2, Comment 7

7. Page 6, Line 133: The subtitle indicating “Method, Research Team, and Reflexivity”. I feel confused why you want to introduce the research team here, especially that only two team members are introduced. It would be clearer for your readers if you could focus on method and reflexivity in this section.

Response: This is the heading that the COREQ guidelines prescribe. We have now added further detail to this section (Editor, Comment X) to provide information about the other researcher’s backgrounds and qualifications, and not just the two researchers who did the primary analysis, as we sometimes report.

Reviewer 2, Comment 8

8. The initial of TP on page 6 is different from the same initial on page 42 under contributions (TRP).

Response: Thank you for pointing this out, we have corrected this now. 

Reviewer 2, Comment 9

9. Page 8, Data analysis section: I would suggest moving the first paragraph of data analysis section to the data collection section.

Response: The information provided in this section is part of the data analysis and we always position it in this section to explain to the reader what the process of interpretation and analysis was. 

Reviewer 2, Comment 10

10. Results: I understand that there is no restriction on word count for submitting to PLOS One, however, the journal does encourage authors to present your findings concisely. I would suggest you to present your result section concisely.

Response: Thank you for this comment. We have indeed tried to balance being concise for readership with providing a platform to share insights from participants in their own words. We have utilised tables to this effect which we believe help strike that balance and have also predominantly limited our quotes to two per example. We hope the editor at PLOS One finds this acceptable but will also make necessary changes to the manuscript as directed. 

Reviewer 2, Comment 11

11. Instead of providing so many quotes, it would be helpful for your readers to understand the lived experience of Asian gbMSM in Australia if you could provide a figure addressing their common experiences, process of mentally transition etc.

Response: Thank you for this response which we have taken into consideration. We do feel the themes that originated from this analysis were more comprehensively displayed in text and Tables as opposed to a figure and this is certainly a format we regularly publish in. We will be happy to revisit this idea at the suggestion of the editor. 

Reviewer 2, Comment 12

12. Page 12: Please provide your rationale why the third theme entitled “Still in a minority group: Experiences of racial discrimination in Australia” cannot be included into the second theme entitled “Living as a gbMSM in Australia”.

Response: We originally saw this as a separate theme since men’s experiences of discrimination were not limited to sexual racism and thus we considered experiences of racism and discrimination as a third theme since it did not always relate to their experience as a gbMSM. Upon reflection, at your suggestion, we have decided to include men’s experiences of racial discrimination in with the second main theme (line 289). 

Reviewer 2, Comment 13

13. Line 239-252. This paragraph sounds to me is an overall conclusion of your research finding. It should be shown at the end of the result section.

Response: In keeping with convention, we have outlined the main findings of our data analysis before introducing the themes individually. We do strongly believe this makes the Results section clearer for the readers and once again, this is a format we regularly use and publish, however, we will revisit this at the discretion of the editor. 

Reviewer 2, Comment 14

14. Line 254. The theme “Life as a gbMSM in country of origin” is really vague to me. Life in a city or a country could be diverse with positivity and negativity. However, the authors only present negative dimensions of experiences toward their sexuality among Asian gbMSM, it could be very misleading and highly biased.

Response: Thank you for this response, please see our previous comment to the editor that addresses this concern as well (Editor, Comment 2) where we explore this aspect of the paper thoroughly and describe appropriate amendments to the text. 

Reviewer 2, Comment 15

15. Page 20, Line 451: The fourth sub-theme “1d. Exposure to HIV-related stigma: ‘HIV is a gay man’s disease’. This theme would be much meaningful and close to your research question if you could spend more space to make the link between HIV stigma and HIV testing in Asia, also parallel to the structure of your second theme (2d). So far, the authors spend more time on describing “HIV is a gay man’s disease”, which is factually and ethically incorrect to statement to make in an academic paper.

Response: Thank you for this comment, we have removed the quote from the title such that the section is called: 1d. Exposure to HIV-related stigma. We understand that you wished to see a stronger connection between the stigmatizing attitudes toward HIV and their experiences of HIV testing in Australia (Section 2d). As for the link between HIV stigma and HIV testing, we have provided clarifying context to the quotes where relevant in the text, as described in response to your additional comment on this topic, Comment 19 below. 

Reviewer 2, Comment 16

16. Page 24, Line 541. “Life as a gbMSM in Australia”- This theme talks about lived experiences of Asian gbMSM in Australia, same as the firs theme, this is really a vague theme for me.

Response: We have rephrased this quote slightly to “Living as a gbMSM in country of origin” which may better capture the essence of the theme, which is to describe men’s experiences of being gbMSM in their countries of origin. We explain in more depth in our response to the Editor’s Comment 2 that many of the themes for this paper are the result of asking men to describe their experience of being a gbMSM in Australia, whereby many men proceeded to describe the differences in acceptance of sexual identity reported in this text. 

Reviewer 2, Comment 17

17. Those quotes related to acceptance and freedom may present part of lived experiences in Australia, other experiences related to racism and xenophobia may cause different experiences. I would suggest the authors to reorganize your themes.

Response: Thank you for this perspective. We have now recategorized Experiences of racial discrimination in Australia as section 2f as we agree that it speaks to part of the experience of living in Australia. Due to the participant’s intersectionality of being gbMSM and also an ethnic minority in Australia, we previously felt it best to hightlight this intersectionality by separating out their experiences of discrimination due to their race from their experiences of being gbMSM in Australia, however we agree that this heading may cause confusion for the reader and have amended it thusly. We still speak to their intersectionality in the discussion, but the theme of discrimination does not need to be separate for this discussion to occur. 

Reviewer 2, Comment 18

18. The findings would be interpreted meaningfully and see the whole picture of Asian gbMSM in Australia if the authors could reorganize your themes beyond personal level perspective. In your quotes, for instance, the authors have found that participants would choose to come out to their friends if the environment is gay friendly and accepting. This information to me is related to interplay between personal, interpersonal, and community level factors. Other quotes also describe social support was associated with coming out among Asian gbMSM.

Response: Thank you for this insight. We have taken this suggestion together with the suggestions from the editor and have bolstered the section 1c. Repressed and hidden identities on men’s experiences in their countries of origin of having peer support. While we previously stated that some men were open to friends but not family members, as you pointed out, we did not elaborate on those men’s experiences. We agree that given the importance of social connections it is best to expand upon this topic as we have now done, please see our response to the editor for Comment 3. 

Reviewer 2, Comment 19

19. Page 30: “2d. Ingrained fear around HIV testing” Please review your quotes under this section carefully and check whether if there is a connection between sociocultural environment and fear of getting an HIV test. The current quotes for me do not support your statement on “many described residual or ingrained fear of HIV testing in Australia as a result of the sociocultural environment in which they grew up, where HIV was heavily stigmatised.” Instead, All the quotes presented in this section are describing their worries and feelings of getting an HIV test. Are those worries a result of that they experienced in their home countries? It would be clear to your reader if you could carefully explain what you concluded.

Response: Thank you for this insight. We have provided more context for the quotes that we have included to better support the connection between delaying testing and fearing HIV due to being exposed to stigmatising attitudes towards HIV.

“A few participants even described delaying testing due to fear of finding out they were HIV positive and what it would mean for their lives. One participant, who had a close circle of friends in his country of origin described how difficult it would be to open up to them about an HIV diagnosis, let alone to his family to whom he was not comfortable being open with his sexual identity. To him, delaying an HIV test was a result of the anxiety he had about what a positive result would mean for his life. 

Yes, I think anxiety might have played a part [in delaying HIV testing] as well…Like I don’t know, I guess it’s better not to find out the bad news if you never get yourself to know; I guess that’s why… I’m not sure if I can ever tell my family about it [an HIV diagnosis]…They would want to know how I contracted HIV and I’m like not even out to them…Yeah, so I don’t think I could ever bring myself to tell my parents especially. I guess my social circle would be quite understanding but I don’t think I would be that courageous to tell them ‘oh I have HIV’. I would have to like build up courage slowly and tell them about my situation now. It would definitely put my, how do you say, my focus and my courage levels would go down… Right now I still have struggles with anxiety like day to day life and all and this would really put like a huge impact. I would be a total mess I would reckon, yes.

—Participant 16, Singapore, 3months in Australia

One participant had his first experience of chemsex (i.e. use of recreational drugs during or before sex) in Australia during which he had sex without a condom and described the fear and anxiety he felt about getting an HIV test in the following weeks. He described feeling like he was HIV positive after the night of chemsex but was too scared to get tested because of what an HIV diagnosis would mean for his life, particularly the isolation he feared he would face living with HIV in his country of origin. 

When the whole drug thing happened with me, the whole five weeks, every single day I was thinking, ‘When do I go? When do I go? Why am I not going? What are you doing with yourself? Why not? Why not?’ I actually thought I was positive….[Having HIV] it would impact in a lot of ways… Now that I’m positive, who’s going to be with me? …[HIV is seen as] Untouchable… Not a lot of people would know about it. Only a close set of people would know about it. I don’t even know if I would tell my parents if I got positive because they’d just break down and they would be – I don’t know. I’ve thought about it a lot of times. I don’t have it but you never know.” 

Reviewer 2, Comment 20

20. Also, quotes in this section actually addressed a very good point but is not developed further by the authors: The reason why they delayed having an HIV test was because they may not know where they can be tested and they don’t know if the sexual health clinics are friendly to racial minorities at the beginning. This statement/point is one of the potential findings that you could explore more.

Response: Thank you for this comment, we do feel we have represented this theme with the quotes provided but will revisit this at the editor’s suggestion.

Reviewer 2, Comment 21

21. Page 33: Theme 3 is a very interesting topic and could contribute to the current literature on the topic of racial discrimination among gbMSM. I would suggest authors to independently produce a manuscript for their experiences.

Response: Thank you for this suggestion. We feel that men’s experiences of racial discrimination tie in with their life as a gbMSM in Australia and thus it makes more sense to include them altogether rather than in a separate manuscript. 

Reviewer 2, Comment 22

22. Discussion, Page 39: The discussion section would be much meaningful if you could provide implications for sexual health practice based on your research findings.

Response: Our separate paper from this study explores facilitators to sexual health, and we have addressed this in the Discussion more clearly with this revision, including implications for improving sexual health in this group (see lines 1113-1123). 

Reviewer 2, Comment 23

23. Line 920-923: “Many men in this study expressed continued anxiety about getting tested for HIV after arriving in Australia due to the negative experiences and HIV-related stigma pervasive in their countries of origin and for some men this resulted in delayed HIV testing” It would be clear to your readers if you make such connect/statement based on your research finding. To me, I did not see the reason of delayed HIV testing was associated with stigma pervasive in their countries of origin. To my understanding, Asian gbMSM in Melbourne may be lacking of HIV testing information to access services.

Response: Thank you for this response, the additional amendments we made to the HIV testing section following your prompt in Comment 19 should make the link clearer between men’s attitudes toward HIV and delaying testing. 

Reviewer 2, Comment 24

24. Line 938-950. If the authors would be interested in writing another paper on the topic of lived experienced of racial discrimination among Asian gbMSM, this paragraph should be removed.

Response: Please see our response above in relation to writing another paper.

Reviewer 2, Comment 25

25. In limitation, authors address that this paper “was not to provide a generalizable findins applicable to all gay or bisexual men born in Asia and living in Australia”. However, you did try to generalize your participants’ experiences in Melbourne to Australia throughout the paper. This issue would be improved if you could switch those parts using “in Australia” to “in Melbourne” to decrease your underlying generalization.

Response: Qualitative research is never intended to be generalisable to a wider population of people - in general, the point of qualitative studies is to explore issues and describe the range of experiences, which we have endeavoured to do. We would not normally be so specific throughout a qualitative manuscript as to where the participants are from as we do not purport to be presenting generalisable data, rather a breadth and depth of experience. We are reluctant to change these parts to “in Melbourne” (and also a few respondents spoke to their experiences in other parts of Australia) however, will do so if the editor feels it is more appropriate. Instead, we have made the following revision to the sentence noted above in the limitations section.

 “…was not to provide a generalizable finding applicable to all gay or bisexual men born in Asia and living in different geographic locations throughout Australia”.

---

## [Decision Letter · Decision Letter 1]

13 Oct 2020

PONE-D-20-17766R1

’Moving from one environment to another, it doesn’t automatically change everything.’ Exploring the transnational experience of Asian-born gay and bisexual men who have sex with men newly arrived in Australia

PLOS ONE

Dear Dr. Phillips,

Thank you for submitting your manuscript to PLOS ONE. You have made considerable revisions in response to the reviewers' and editor's comments, and the manuscript is much improved. After careful consideration, we feel that it has merit but does not fully meet PLOS ONE’s publication criteria as it currently stands. For one, PLOS ONE does not copy edit manuscripts, so the following corrections need to be made. Additionally, a few statements need to be revised. Therefore, we invite you to submit a revised version of the manuscript that addresses the points raised during the review process (line numbers refer to revised version with track changes):

Line 221: Should be 2e. it jumps from 2d. to 2f.

Line 916: Should be: Skills 2e.

1080: You refer to a “separate paper” with no reference. At least you should reference this as an “unpublished manuscript”. However, given it has not completed peer-review, you should also cite published articles that address social support among GBMSM, of which there are many:

See for ex. an earlier systematic review that includes MSM in the context of HIV risk, & others:

Qiao S, Li X, Stanton B. Social support and HIV-related risk behaviors: a systematic review of the global literature. *AIDS Behav*. 2014;18(2):419-441. doi:10.1007/s10461-013-0561-6

Saleh LD, van den Berg JJ, Chambers CS, Operario D. Social support, psychological vulnerability, and HIV risk among African American men who have sex with men. *Psychol Health*. 2016;31(5):549-564. doi:10.1080/08870446.2015.1120301

1082: fix the apostrophe. Change to participants’

1157: fix the apostrophe. Change to country’s

1161: This is qualitative research. You didn’t “measure” anything, so this statement sounds strange. In another place in the manuscript you similarly note that you did not "measure resilience". These statements should be revised as quantitaive assessment is not generally part of qualitative research; it is not specific to your study.

1220-1222: You cite one source that apparently supports the claim that the use of more than one coder is “generally prohibitive” due to “cost and effort”: “While ideal, this is generally prohibitive in qualitative research due to cost and effort, when compared with the standard convention of cross-checking by a researcher in a supervisory role [44].”

This statement is misleading and overstated as written. The use of one coder is not the norm for qualitative analysis (and there are many references available to this end; for ex., see below). Indeed this is why the specific item is included on the COREQ checklist (see No. 24). Generally two to three independent coders are recommended. You need to revise this statement to correctly indicate this, although *you *may have used only one coder due to concerns about “cost and effort” and that shortcoming may be characteristic of some qualitative research (“particularly in early-career contexts” [Campbell et al., 2013]). Another source indicates the importance of carrying reflexivity through the data analytic stage, which may help to mitigate (though not erase) bias.

See for ex.:

Campbell, J. L., Quincy, C., Osserman, J., Pedersen, O. K. (2013). Coding in-depth semistructured interviews: Problems of unitization and intercoder reliability and agreement. Sociological Methods & Research, 42, 294–320. https://doi.org/10.1177/0049124113500475

Church, Sarah, Michael Dunn, and Linda Prokopy. 2019. "Benefits to Qualitative Data Quality with Multiple Coders: Two Case Studies in Multi-coder Data Analysis." Journal of Rural Social Sciences, 34(1): Article 2. Available At: https://egrove.olemiss.edu/jrss/vol34/iss1/2

Lacy, Stephen; Watson, Brendan R.; Riffe, Daniel; and Lovejoy, Jennette, "Issues and Best Practices in Content Analysis" (2015). Communication Studies Faculty Publications and Presentations. 8. http://pilotscholars.up.edu/cst_facpubs/8

Tufford L, Newman P. Bracketing in Qualitative Research. *Qualitative Social Work*. 2012;11(1):80-96. doi:10.1177/1473325010368316

[e.g., Walther et al. (2013) suggested IRR as a means to “mitigate interpretative bias” and ensure a “continuous dialogue between researchers to maintain consistency of the coding” (p. 650). Miles and Huberman (1994) suggest that an IRR of 80% agreement between coders on 95% of the codes is sufficient agreement among multiple coders(Miles & Huberman, 1994)]

We look forward to receiving your revised manuscript.

Kind regards,

Peter A Newman, Ph.D

Academic Editor

PLOS ONE

Reviewers' comments:

Reviewer's Responses to Questions

**Comments to the Author**

1. If the authors have adequately addressed your comments raised in a previous round of review and you feel that this manuscript is now acceptable for publication, you may indicate that here to bypass the “Comments to the Author” section, enter your conflict of interest statement in the “Confidential to Editor” section, and submit your "Accept" recommendation.

Reviewer #1: All comments have been addressed

2. Is the manuscript technically sound, and do the data support the conclusions?

Reviewer #1: Yes

3. Has the statistical analysis been performed appropriately and rigorously? 

Reviewer #1: (No Response)

4. Have the authors made all data underlying the findings in their manuscript fully available?

Reviewer #1: (No Response)

5. Is the manuscript presented in an intelligible fashion and written in standard English?

Reviewer #1: Yes

6. Review Comments to the Author

Reviewer #1: (No Response)

7. PLOS authors have the option to publish the peer review history of their article (what does this mean?). If published, this will include your full peer review and any attached files.

Reviewer #1: No

---

## [Author Response · Author response to Decision Letter 1]

28 Oct 2020

Editorial Requests and Comments

Thank you for submitting your manuscript to PLOS ONE. You have made considerable revisions in response to the reviewers' and editor's comments, and the manuscript is much improved. After careful consideration, we feel that it has merit but does not fully meet PLOS ONE’s publication criteria as it currently stands. For one, PLOS ONE does not copy edit manuscripts, so the following corrections need to be made. Additionally, a few statements need to be revised. Therefore, we invite you to submit a revised version of the manuscript that addresses the points raised during the review process (line numbers refer to revised version with track changes):

1. Line 221: Should be 2e. it jumps from 2d. to 2f.

Response: Thank you, this has been changed

2. Line 916: Should be: Skills 2e.

Response: Thank you, this has been changed

3. 1080: You refer to a “separate paper” with no reference. At least you should reference this as an “unpublished manuscript”. However, given it has not completed peer-review, you should also cite published articles that address social support among GBMSM, of which there are many:

See for ex. an earlier systematic review that includes MSM in the context of HIV risk, & others:

Qiao S, Li X, Stanton B. Social support and HIV-related risk behaviors: a systematic review of the global literature. AIDS Behav. 2014;18(2):419-441. doi:10.1007/s10461-013-0561-6

Saleh LD, van den Berg JJ, Chambers CS, Operario D. Social support, psychological vulnerability, and HIV risk among African American men who have sex with men. Psychol Health. 2016;31(5):549-564. doi:10.1080/08870446.2015.1120301

Response: Thank you for this advice. Where our ‘separate’ publication is mentioned we add that the manuscript is pending publication. For example,

Line 109: “The overall aim of this study was to explore HIV knowledge and prevention strategies used and preferred among newly-arrived Asian-born gbMSM, the results of which are pending publication in a separate paper”

Line 248: “Findings related to men’s HIV and STI knowledge and HIV prevention strategies are reported in a separate paper (as yet unpublished).”

Line 1117: “In our separate paper from this study (as yet unpublished), we discuss our findings that social support was a facilitator to sexual health within this group…”

Additionaly, in line with your comment about including published references for this citation we have amended the text, adding the suggested references:

“In our separate paper from this study (unpublished manuscript), we discuss our findings that social support was a facilitator to sexual health within this group, notably some participants’ positive experiences with local LGBTQI organisations that offer peer-led social support workshops. Previous research has also found that social support can impact sexual behaviours (1, 2). Increasing awareness of and opportunities to engage with peer-led services may be beneficial for men in this group in terms of bolstering their social support in Australia.”

4. 1082: fix the apostrophe. Change to participants’

Response: Changed

5. 1157: fix the apostrophe. Change to country’s

Response: Changed

6. 1161: This is qualitative research. You didn’t “measure” anything, so this statement sounds strange. In another place in the manuscript you similarly note that you did not "measure resilience". These statements should be revised as quantitaive assessment is not generally part of qualitative research; it is not specific to your study.

Response: Thank you for your comment. We have amended the text as below:

Line 1147: “While measuring resilience (i.e. the ability to recover after experiencing adversity) among our participants was outside the scope of this study, peer support is a known protective process that leads to the development of resilience”

And 

Line 1232: “This study did not ask participants to describe the degree to which they identified with a particular ethnicity, but recent research has questioned the role of ethnic identity as a protective buffer for the stress of discrimination (3). Future research could investigate whether men in this group with stronger ethnic identity have ameliorated stress levels in the face of discrimination.“

7. 1220-1222: You cite one source that apparently supports the claim that the use of more than one coder is “generally prohibitive” due to “cost and effort”: “While ideal, this is generally prohibitive in qualitative research due to cost and effort, when compared with the standard convention of cross-checking by a researcher in a supervisory role [44].”

This statement is misleading and overstated as written. The use of one coder is not the norm for qualitative analysis (and there are many references available to this end; for ex., see below). Indeed this is why the specific item is included on the COREQ checklist (see No. 24). Generally two to three independent coders are recommended. You need to revise this statement to correctly indicate this, although you may have used only one coder due to concerns about “cost and effort” and that shortcoming may be characteristic of some qualitative research (“particularly in early-career contexts” [Campbell et al., 2013]). Another source indicates the importance of carrying reflexivity through the data analytic stage, which may help to mitigate (though not erase) bias.

See for ex.:

Campbell, J. L., Quincy, C., Osserman, J., Pedersen, O. K. (2013). Coding in-depth semistructured interviews: Problems of unitization and intercoder reliability and agreement. Sociological Methods & Research, 42, 294–320. https://doi.org/10.1177/0049124113500475

Church, Sarah, Michael Dunn, and Linda Prokopy. 2019. "Benefits to Qualitative Data Quality with Multiple Coders: Two Case Studies in Multi-coder Data Analysis." Journal of Rural Social Sciences, 34(1): Article 2. Available At: https://egrove.olemiss.edu/jrss/vol34/iss1/2

Lacy, Stephen; Watson, Brendan R.; Riffe, Daniel; and Lovejoy, Jennette, "Issues and Best Practices in Content Analysis" (2015). Communication Studies Faculty Publications and Presentations. 8. http://pilotscholars.up.edu/cst_facpubs/8

Tufford L, Newman P. Bracketing in Qualitative Research. Qualitative Social Work. 2012;11(1):80-96. doi:10.1177/1473325010368316

[e.g., Walther et al. (2013) suggested IRR as a means to “mitigate interpretative bias” and ensure a “continuous dialogue between researchers to maintain consistency of the coding” (p. 650). Miles and Huberman (1994) suggest that an IRR of 80% agreement between coders on 95% of the codes is sufficient agreement among multiple coders(Miles & Huberman, 1994)]

Reponse: Thanks for this insight. The reference that we cited in our manuscript in the line you are referring to (Barbour, 2001: https://www.ncbi.nlm.nih.gov/pubmed/11337448) does indeed caution against multiple people coding the entirety of the data, here is the excerpt from the text: “While I would caution against multiple coding of entire datasets (on the grounds of economy in both cost and effort), some element of multiple coding can be a valuable strategy. It can be useful to have another person cast an eye over segments of data or emergent coding frameworks, and this is a core activity of supervision sessions and research team meetings.” 

We have now changed the text to remove the suggestion that multiple coders are not the norm:

Additionally, had two to three researchers coded the entirety of the data (cross-coding/multiple coding) instead of one researcher doing the majority of the analysis, this would have provided more rigor to the analysis. While it would have been ideal to undertake cross coding of the data using multiple coders, which would have further assisted in mitigating bias, this was not possible due to time and resource limitations. This approach serves as a valuable strategy where these limitations apply (4).

1. Qiao S, Li X, Stanton B. Social support and HIV-related risk behaviors: a systematic review of the global literature. AIDS Behav. 2014;18(2):419-41.

2. Saleh LD, van den Berg JJ, Chambers CS, Operario D. Social support, psychological vulnerability, and HIV risk among African American men who have sex with men. Psychol Health. 2016;31(5):549-64.

3. Mossakowski KN, Wongkaren T, Hill TD, Johnson R. Does ethnic identity buffer or intensify the stress of discrimination among the foreign born and U.S. born? Evidence from the Miami-Dade Health Survey. J Community Psychol. 2019;47(3):445-61.

4. Barbour RS. Checklists for improving rigour in qualitative research: a case of the tail wagging the dog? BMJ. 2001;322(7294):1115-7.

---

## [Editor Report · Decision Letter 2]

10 Nov 2020

’Moving from one environment to another, it doesn’t automatically change everything.’ Exploring the transnational experience of Asian-born gay and bisexual men who have sex with men newly arrived in Australia

PONE-D-20-17766R2

Dear Dr. Phillips,

We’re pleased to inform you that your manuscript has been judged scientifically suitable for publication and will be formally accepted for publication once it meets all outstanding technical requirements.

Kind regards,

Peter A Newman, Ph.D

Academic Editor

PLOS ONE
---

## [Editor Report · Acceptance letter]

12 Nov 2020

PONE-D-20-17766R2 

“Moving from one environment to another, it doesn’t automatically change everything.” Exploring the transnational experience of Asian-born gay and bisexual men who have sex with men newly arrived in Australia 

Dear Dr. Phillips:

I'm pleased to inform you that your manuscript has been deemed suitable for publication in PLOS ONE. Congratulations! Your manuscript is now with our production department. 

Kind regards, 

on behalf of

Dr. Peter A Newman 

Academic Editor

PLOS ONE